# Rydberg electron stabilizes the charge localized state of the diamine cation

**Marc Reimann** [1], **Christoph Kirsch** [2], **Daniel Sebastiani** [2] **& Martin Kaupp** [1] ✉

A previous controversial discussion regarding the interpretation of Rydberg spectra of gaseous dimethylpiperazine (DMP) as showing the co-existence of a localized and delocalized mixed-valent DMP⁺ radical cation is revisited. Here we show by high-level quantum-chemical calculations that an apparent barrier separating localized and delocalized DMP⁺ minima in previous multi-reference configuration-interaction (MRCI) calculations and in some other previous computations were due to unphysical curve crossings of the reference wave functions. These discontinuities on the surface are removed in state-averaged MRCI calculations and with some other, orthogonal high-level approaches, which do not provide a barrier and thus no localized minimum. We then proceed to show that in the actually observed Rydberg state of neutral DMP the 3s-type Rydberg electron binds more strongly to a localized positive charge distribution, generating a localized DMP* Rydberg-state minimum, which is absent for the DMP⁺ cation. This work presents a case where interactions of a Rydberg electron with the underlying cationic core alter molecular structure in a fundamental way.

Mixed-valence (MV) compounds are important models for a large variety of electron transfer (ET) processes, many are also important in their own right for various uses. Therefore, MV systems have been widely studied for many decades, both experimentally and by means of (semi-)empirical theoretical models and explicit quantum-chemical computations. The literature on this topic is too vast to do justice here, and we refer the reader to some reviews and books[1–4]. Almost all spectroscopic studies are carried out in polar solvent environments, due to the fact that MV systems are often charged (radical) species that do not dissolve readily in nonpolar solvents. Such polar solvents, however, favor charge localization due to electrostatic interactions. Thus, they may often influence molecular and electronic structure, and the solvent should be viewed as an intrinsic part of the MV system in question[1]. As this complicates a deeper understanding of the electronic structure, and particularly any quantitative computational modeling, systems that may be studied in the gas phase are of particular interest. However, not many MV gas-phase systems have become accessible experimentally so far. One of these few, the gaseous dimethylpiperazine

radical cation (DMP⁺), has generated particular interest in the recent past.

While DMP⁺ has been known in solution for some time[5,6], its gas-phase existence and structure has been inferred from Rydberg spectroscopy of neutral gaseous DMP[7]. In these experiments, an electron from one of the two nitrogen lone-pairs is excited into an orbital with nitrogen 3p Rydberg character. Within a few hundred fs, this state relaxes to give a Rydberg state with nitrogen 3s character. The process is followed by tracing the remaining binding energy of the Rydberg electron with the cationic, molecular core. As the Rydberg electron usually has a small effect on the molecular structure of the core, this Rydberg state has been assumed to behave identically to the respective cation and was therefore modeled as such[7–9].

The substantial interest generated by this work originates from the apparent presence of two distinct molecular structures in fast equilibrium: upon excitation, the 3p-type Rydberg state is converted to an initial 3s-type state exhibiting a broad signal with a time constant of about 500 fs, decaying toward a sharper signal over a few ps[7,9]. The observations were interpreted as the initial formation of a charge-

[1]Theoretische Chemie/Quantenchemie, Institut für Chemie, Sekr. C7, Technische Universität Berlin, 10623 Berlin, Germany. [2]Institut für Chemie, Martin-Luther-University Halle-Wittenberg, 06120 Halle (Saale), Germany. ✉e-mail: martin.kaupp@tu-berlin.de

localized cation state (DMP-L$^+$), where the electron hole is largely confined to one amine center. This then decays in part to the more stable charge-delocalized state (DMP-D$^+$) known also in solution[5,6]. The energy difference between these two states was determined to be 0.33 ± 0.04 eV[8]. Such a simultaneous existence of a localized and a delocalized structure of an MV radical cation is uncommon, although a coexistence of localized and delocalized conformers is known for more complex organometallic MV systems in solution (see ref. 1 for a review).

Notably, most quantum-chemical methods did not agree with the interpretation of coexisting structures of DMP$^+$, as they did not find a barrier between the two states, i.e., they did not obtain DMP-L$^+$ as a distinct minimum on the potential energy surface (PES)[8]. This holds for most exchange-correlation functionals of Kohn–Sham DFT, except for the somewhat non-standard BHandHLYP functional with 50% exact exchange (EXX) admixture[10] and the so-called Perdew-Zunger self-interaction correction (PZ-SIC)[11]. Criticizing the conclusion of ref. 8 that most DFT levels are in error for this system, it has been pointed out that even coupled-cluster theory at the CCSD(T) "gold-standard" level does not give a barrier (while unrestricted CCSD gives a very small one)[12]. In a reply this has been rejected as a shortcoming of single-reference CCSD(T) due to multi-reference character of the wave function at the transition state[13]. Indeed, the typical diagnostics for multi-reference character at the barrier have been interpreted rather differently by these two sets of authors. Most recently, Gałyńska et al.[14] reported multi-reference configuration interaction (MRCI+Q) energy calculations on a set of structures from the BHandHLYP potential-energy surface to support the presence of a barrier and the inadequacy of CCSD(T) and of most density functionals. Based on these data, they considered the DMP$^+$ puzzle solved.

In this work we resolve the controversial discussion of the simultaneous existence of a delocalized and localized mixed-valent (MV) minimum of the prototypical DMP$^+$ cation. Previous Rydberg spectroscopy experiments have been interpreted to show such an existence, while computations suggested that many quantum-chemical methods fail in the description of this situation. This had been criticized as incorrect by other authors. Here we show by high-level quantum-chemical computations that the localized structure exists only on the potential-energy surface of the Rydberg state but not on the surface of the fully ionized state. In addition to showing shortcomings of earlier calculations, our results reveal an unusually large effect of a Rydberg electron on a molecular structure.

## Results and discussion
### A comment on previous literature
This controversy intrigued us for various reasons: (1) several of the DFT functionals tested and suggested to fail have provided a good balance between localized and delocalized MV situations in previous work[1,2], while BHandHLYP clearly was biased toward too localized structures (see, e.g., evaluations for the MVO-10 gas-phase oxide MV benchmark set[15]). PZ-SIC has also been known to overcorrect and deteriorate results in a number of cases (see, e.g., refs. 16,17 and cited literature). Even M06-HF[18], which also did not give a barrier, is known to be clearly biased toward localized states. (2) Only DMP-D$^+$ could be identified in solution experiments in acetonitrile, with no indication for the existence of a localized isomer[5,6]. This seems at odds with the gas-phase existence of DMP-L$^+$, as polar solvents are known to stabilize localized MV states (see above). (3) The reported multi-reference diagnostics[14] (in particular the natural orbital occupation numbers, NOON) did indeed not strike us as indicative of substantial multi-reference character. In fact, NOONs of 0.01 or 1.99 indicate to us that the active space might already be significantly larger than necessary. Moreover, if a lack of static correlation at the transition state were the issue for the CCSD(T) calculations, one would expect this method to overestimate the barrier rather than underestimate it, as static correlation for

DMP-L$^+$ and DMP-D$^+$ should be less pronounced than for the transition state. Notably, a good performance of BHandHLYP or PZ-SIC also does not seem to be consistent with appreciable multi-reference character, as large EXX admixtures and PZ-SIC typically worsen the description of static correlation[16,17,19].

### Investigation of the DMP cation
We therefore decided to revisit this problem, using various different quantum chemical tools. As the two-dimensional potential energy surfaces shown in ref. 14 have been built from a rather small number of points in a given direction (78 points in total, roughly 9 for each direction), we chose to generate a one-dimensional effective reaction path with a total of 25 points between the optimized structures of DMP-D$^+$ (point 1) and DMP-L$^+$ (point 25) at the BHandHLYP level (see "Methods" section for details). As these points are no longer trivially connected to any specific internal coordinate, this provides a reasonable description of the interesting parts of the potential energy surface with a significantly lower number of points, allowing various high-level methods to be probed more efficiently.

Using this pre-generated reaction path, we revisited many of the previously employed single-reference methods (see Supplementary Discussion 1 for a detailed exposition). A number of conclusions emerged from these comparisons: (1) A barrier for DMP$^+$ at UHF level is an artifact of spatial symmetry breaking in the absence of Coulomb correlation (it is present also at ROHF level). The barrier is then caused by a curve crossing of two qualitatively different localized solutions. (2) This is "passed down" to a BHandHLYP barrier due to large EXX admixture. (3) A small CCSD barrier is also due to an incomplete correction of the spatial symmetry breaking. Notably, the barrier is absent at IP-EOM-CCSD level, i.e., when using a closed-shell reference wave function! (4) We find no indication that the perturbative (T) contribution would suffer from any problems with a multi-reference situation[12], the vanishing barrier appears to be physically sound. This is confirmed by various diagnostics for multi-reference character (see Supplementary Discussion 1).

This leaves us of course with the question why the MRCI+Q results of ref. 14 produced a barrier. Performing equivalent calculations we were able to reproduce a barrier of about 0.05 eV on the pre-generated path (dashed orange curve in Fig. 1). Notably, however, the underlying (state-specific) CASSCF reference curve (solid orange, Fig. 1) is not smooth but shows a pronounced cusp at the position of the barrier found at MRCI + Q level. This is in turn accompanied by an abrupt change in the nature of the wave function as indicated by the change in Mulliken atomic spin density on the nitrogen atom bearing the hole (orange dots). The CASSCF curve follows qualitatively the switch of the lowest-energy HF solution at the same points (cf. Supplementary Figs. 1 and 2), with the wrong relative stability of the two structures. The barrier at the state-specific MRCI+Q level thus has the same origin as the CCSD barrier: single and double substitutions (out of a CASSCF or HF reference, respectively) are insufficient to correct the artificial spatial symmetry breaking of the reference wave function, and the resulting curve exhibits the remainders of this symmetry breaking. The similarities of the HF and CASSCF curves reflect the fact that the leading determinant of the CASSCF has a CI coefficient of more than 0.96 throughout the reaction path. Together with the rather small deviation of the natural orbital occupation numbers from the idealized case (minimum NOON for formally occupied orbitals 1.96, maximum NOON for formally virtual orbitals 0.02), this implies that the CASSCF treatment does not significantly improve upon HF, i.e., not much static correlation is generated. Additional calculations using larger active spaces (see Supplementary Discussion 2) add some dynamic correlation and therefore give somewhat smoother surfaces, but they do not alter the basic conclusions on the underlying symmetry breaking, although they improve the relative stability of the two minima. A much smoother change in orbital occupations and spin density is obtained

by performing state-averaged calculations including the ground and first excited doublet states (with equal weights, teal colored dots). Interestingly, this does not only smooth the energy curve, it also leads to a qualitatively correct relative energy between the DMP-D$^+$ and DMP-L$^+$ regions (teal colored dashed and solid curves; some results obtained for non-equal weights are shown in Supplementary Fig. 12). We note in passing, however, that with any currently accessible active space and one-particle basis set a CASSCF or DMRG calculation will give only a small fraction of dynamical correlation energies and therefore cannot be viewed as a benchmark computational level.

Most importantly, the shape of the MRCI+Q curve based on state-averaged CASSCF orbitals agrees very closely with those obtained with spin-unrestricted CCSD(T) (using either HF, Kohn–Sham or Brueckner orbitals) and spin-restricted IP-EOM-CCSD theory (see Supplementary Figs. 5, 6 and 8). We also find that the shape of the state-averaged CASSCF curve agrees with that obtained from a non-orthogonal CI based on the three different UHF solutions discussed in Supporting Information (see Supplementary Fig. 10). We note in passing that inclusion of the second excited doublet state in the state averaging does not affect these findings (see Supplementary Fig. 13).

Based on consistent analyses at a wide variety of computational levels, we can thus clearly state that it is extremely unlikely that DMP$^+$ exhibits a local DMP-L$^+$-type minimum! However, this leaves us still with the apparent contradiction with respect to the interpretation of the time-resolved Rydberg spectra of DMP in terms of a delocalized and a localized cation state[7]. We initially thought the barrier might be entropic in nature and therefore carried out ab initio molecular dynamics simulations on DMP$^+$ at the PBE0/DZVP-MOLOPT-GTH level. While DMP-D$^+$ shows molecular vibrations resulting in structures with planar nitrogen atoms found in the pre-generated reaction path, DMP-L$^+$ is not observed and thus no indications for an entropic barrier are found. For more details on the molecular dynamics simulations, we refer the reader to Supplementary Discussion 3. We note in passing

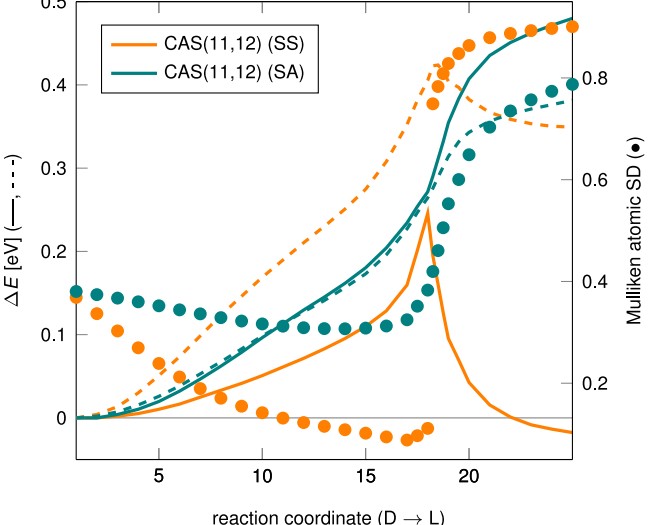

**Fig. 1 | Multi-reference results for the dimethylpiperazine (DMP) cation.** Complete active space (CAS) self-consistent field (CASSCF, solid) and multi-reference configuration interaction with Davidson correction (MRCI+Q, dashed) energies in eV relative to the delocalized minimum along the pre-generated pathway from the delocalized (D) to the localized (L) structure. Either state-specific (SS, orange) or state-averaged (SA, teal colored; see text for details) calculations are shown. CASSCF Mulliken spin densities at the hole-bearing nitrogen atom in the localized structure are given as dots. A CAS(11,12) space as in ref. 14 (see "Methods" section) and aug-cc-pVDZ basis sets were used. Source data are provided as a Source data file.

that a coherent-dynamics rather than kinetic origin of the spectral evolution has been ruled out[13].

## Investigation of the DMP Rydberg state

A solution to this conundrum can be found by considering more closely the nature of the actual experiment. The time evolution of a 3s-type Rydberg state is observed, not that of a radical cation. Calculating the corresponding excitation energies at the EOM-CCSD/aug-cc-pVDZ level of theory using a BHandHLYP/def2-TZVPPD optimized structure for the DMP $C_{2h}$-symmetric ground-state structure shows 8 excitation energies below 6.5 eV which can all be classified as valence-to-Rydberg excitations from one of the two nitrogen lone pairs to the 3s and 3p Rydberg orbitals. This does not only match the experimentally used energy of the pump excitation (207 nm, 5.99 eV)[7], it also makes the Rydberg state of interest the first excited state of the neutral DMP! This reflects the saturated nature of the molecule and the absence of low-lying $\pi^*$ orbitals. Indeed, the remaining binding energy of the electron in the lowest-lying Rydberg state is ca. 2.5 eV at the IP-EOM-CCSD/aug-cc-pVDZ level. This value indicates appreciable binding and is large compared to the energy difference between charge-localized and delocalized cation states of about 0.3 eV. This suggests that the interaction between the Rydberg electron and the residual charged core may have a non-negligible influence on the potential-energy surface of the Rydberg state.

To model the decisive region of the potential-energy surface of the Rydberg state in more detail, we resorted to the LR-SCS-CC2 level, i.e., to linear-response theory with spin-component-scaled CC2 energies (which is more computationally efficient but otherwise similar to EOM-CCSD[20]), based on a closed-shell RHF reference. Computing a relaxed reaction path at this level, we do indeed find the suggested charge-localized minimum structure DMP-L$^*$ and a barrier connecting it to a charge-delocalized minimum DMP-D$^*$ that corresponds closely to the structure of DMP-D$^+$. Very similar potential energy surfaces can also be obtained at the linear-response time-dependent DFT (TDDFT) level using various functionals. Reduced self-interaction errors with range-separated hybrid functionals like $\omega$B97X-D move the relative energies of the two minima closer to the LR-SCS-CC2 value and increase somewhat the barrier (Supplementary Table 1), making it more consistent with the relatively rough experimental estimate of ca. 0.1 eV[13]. Most density functionals evaluated are also reasonably well suited to calculate the binding energy of the Rydberg electron (see Supplementary Table 2). LR-SCS-CC2 bond lengths are in reasonable agreement with recent experimental values for the Rydberg state (see Supplementary Table 3), within the accuracy afforded by the underlying femtosecond X-ray scattering experiments[9].

To investigate the origin of this qualitative difference between the Rydberg state and the cation, we computed (EOM-)CCSD energies for the electronic ground state of neutral DMP, for the Rydberg state, and for the ionized state on the LR-SCS-CC2/aug-cc-pVTZ reaction path (Fig. 2). Our approach differs from the EOM-CCSD-based calculations presented in ref. 8, that employed MP2-optimized structures of the cation and calculated the binding energy of the Rydberg electron from the vertical ionization energy at CCSD(T) level supplemented with an EOM-CCSD excitation energy. Here, we use the IP-EOM-CCSD approach to directly compare the energies of cation and Rydberg state, which eliminates any possible errors originating from a UHF reference for the cation.

Starting from the delocalized minimum, the energies of the Rydberg and the ionized state develop in a closely parallel fashion until the transition state on the Rydberg surface is reached. After this point, the binding energy of the Rydberg electron increases from around 2.54 eV to about 2.65 eV. This difference originates from a larger Coulombic stabilization of the localized DMP-L$^+$-type core by the Rydberg electron compared to the delocalized DMP-D$^+$-type core. The change in binding energies of the Rydberg electron also matches the observed

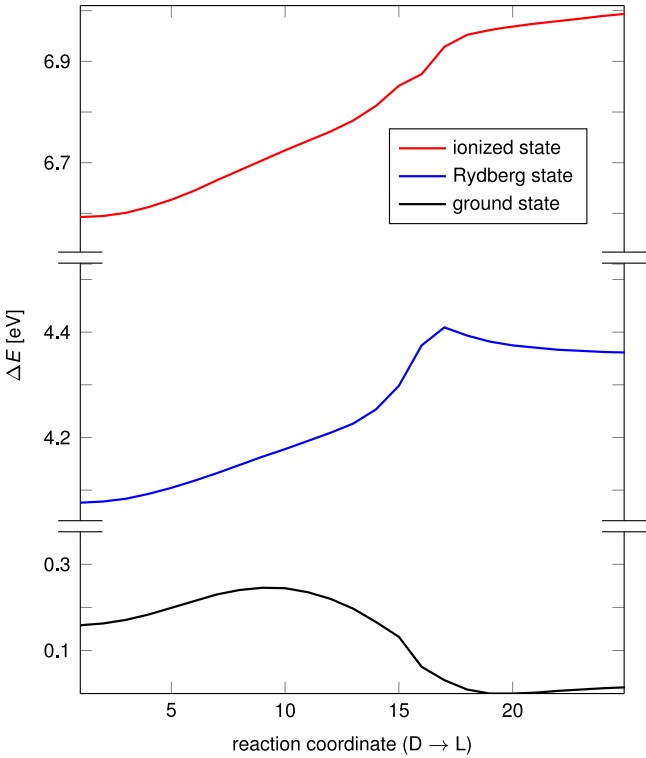

**Fig. 2 | Energy surfaces of the dimethylpiperazine (DMP) system.** Relative energies in eV for neutral, Rydberg and ionized states of DMP calculated at coupled cluster level with single and double excitations (CCSD), equations-of-motion (EOM-)CCSD level and ionization potential (IP-)EOM-CCSD level, respectively, along the reaction path for the Rydberg state from the delocalized (D) to the localized (L) structure optimized at the linear-response spin-component scaled coupled cluster (LR-SCS-CC2) level. Source data are provided as a Source data file.

experimental increase of about 0.11 eV from the delocalized to the localized state[7]. The relatively large changes in binding energy on the right hand side of the (rather late) transition state may also provide an explanation for the appreciably broader signal found for the localized compared to the delocalized state. In this context we note that the atomic motion along the path in this DMP-L* region consists mostly in a planarization of the hole-carrying nitrogen atom. We also note in passing that this enhanced interaction between Rydberg electron and a localized cationic core is reminiscent of the enhanced stabilization of localized MV ions by polar solvents, e.g., in the case of hydrogen bonding to MV radical anions[1–4].

The LR-SCS-CC2/aug-cc-pVTZ energy of DMP-L* relative to DMP-D* (0.24 eV) corresponds very well to the experimental enthalpy difference between the two states of $0.22 \pm 0.04$ eV[8]. The improvement is somewhat reduced at the EOM-CCSD/aug-cc-pVTZ level (0.29 eV), hinting at error compensation effects in the SCS-CC2 treatment. The underestimate of the binding energy of the Rydberg electron by about 0.15 eV at the IP-EOM-CCSD/aug-cc-pVTZ level may reflect the lack of higher excitations. In any case, both LR-SCS-CC2/aug-cc-pVTZ and IP-EOM-CCSD/aug-cc-pVTZ calculations are thought to provide an accurate description of the potential-energy surface of DMP*.

In conclusion, we have shown here that the previously suggested localized form of the DMP$^+$ mixed-valence radical cation computed at multi-reference CI level is the result of artificial spatial symmetry breaking and curve crossings. When eliminating these artifacts by state-averaged calculations, a smooth potential-energy surface without barrier or a localized minimum is obtained. This supports earlier results at CCSD(T) level and by most DFT functionals. Artificial barriers obtained previously at BHandHLYP or PZ-SIC levels mirror an unphysical symmetry breaking observed also in UHF calculations. These

definite computational results for the DMP$^+$ cation are at variance with the interpretation of Rydberg spectroscopy experiments on neutral DMP, and we also could not corroborate an explanation via entropic barriers. We have therefore taken the extra step of modeling explicitly the potential-energy surface of the excited Rydberg state. It turns out, that due to the saturated nature of DMP this is the lowest excited state, and the nitrogen 3s-type Rydberg electron is rather strongly bound. In fact, the computations at various levels (LR-SCS-CC2, IP-EOM-CCSD, TDDFT with suitable functionals) show that the stronger binding of the Rydberg electron across the localized portions of the potential-energy surface generates a DMP-L* local minimum. That is, this localized mixed-valence state does not exist for the isolated DMP$^+$ cation but only for the mixed-valent Rydberg state with its bound extra electron. This is evidence for a structure-determining role of unusual magnitude, as typical structural effects of Rydberg electrons are much smaller[21]. We suspect that further examples of this type might be found for other Rydberg states of saturated species, where the Rydberg electron tends to be more strongly bound than for unsaturated systems. We also note that, in contrast to previous claims[8,13,14], DMP$^+$ is not an example for a wide failure of DFT or single-reference wave-function methods.

## Methods
### Calculations on DMP$^+$
As one of the main goals of this work is the comparison of various different (high-level) electronic-structure methods regarding the existence of DMP-L$^+$, a reliable description of the potential-energy surface with a small number of points is desirable. As only the BHandHLYP and the PZ-SIC functionals have been found to provide local minima for DMP-L$^+$[8], the structures of DMP-L$^+$ and DMP-D$^+$ were initially optimized at the BHandHLYP level, using the TURBOMOLE program package[22,23], def2-TZVPPD basis sets[24], a fine integration grid (gridsize 5) and D3 atom-additive dispersion corrections with Becke-Johnson damping[25,26]. To speed up the calculations, the multipole-accelerated resolution of the identity (MARIJ) approximation[27] in combination with the appropriate auxiliary basis functions[28] was used. Exact-exchange integrals were solved semi-numerically ($senex)[29] using a moderately sized grid (gridsize 1) and de-aliasing. Both structures were characterized as minima by harmonic vibrational frequency analyses. The woelfling module[30] of TURBOMOLE has been employed to generate a minimum-energy reaction path (containing 25 points) from DMP-D$^+$ to DMP-L$^+$ based on a quadratic PES, using a chain-of-states algorithm[30]. To increase the resolution close to the transition state, 5 additional points were added by linear interpolation between existing structures of the converged reaction path. All obtained structures are given in Supplementary Data 1.

The performance of various different electronic-structure methods was then tested by additional energy calculations at these points along the BHandHLYP reaction path, generally with aug-cc-pVDZ basis sets[31], consistent with ref. 14. Coupled-cluster calculations for DMP$^+$ at the spin-unrestricted level using either UHF or BHandHLYP (GKS) reference functions or Brueckner orbitals were performed with the TURBOMOLE program package using the appropriate auxiliary basis sets[32]. Additional coupled-cluster calculations using ROHF reference functions and spin-unrestricted coupled-cluster equations were performed using the MOLPRO program package in the 2022.2 release[33,34].

Equation-of-motion (EOM) coupled-cluster calculations both in the IP and the EE formulation (for the cation and the Rydberg state, respectively) were performed using the back-transformed pair-natural orbital (bt-PNO) implementation in the ORCA program package[35,36], Version 5.0.2, with adjusted TightPNO settings (TCutPNO was set to $10^{-8}$), VeryTight SCF convergence criteria and the resolution of the identity for both the Coulomb and the exchange matrix. The necessary JK auxiliary basis sets were obtained using the AutoAux procedure[37]. Redundant basis functions were removed using a pivoted Cholesky

decomposition of the Coulomb metric with an automatically determined threshold. All bt-PNO calculations employed a resolution of the identity for the integral transformation using the appropriate auxiliary (aug-cc-pVDZ/C) basis sets and the fully linear scaling implementation (DLPNOLINEAR true and NEWDOMAINS true).

All CASSCF-based calculations on DMP$^+$ were performed using MOLPRO[33,34]. The active space was chosen identically to the active space given in ref. [14], consisting of 11 electrons in 12 orbitals. The active orbitals include the two nitrogen lone-pair orbitals, their respective double shell orbitals, the two C-C bonding orbitals and the two N-Me bonds as well as the respective anti-bonding orbitals. A representation of the final natural orbitals can be found in Supplementary Figs. 16 through 19. Based on this active space, multi-reference configuration interaction (MRCI+Q) calculations were performed using the CW-contracted CISD routine[38]. All energies include Davidson's +Q size-consistency correction[39]. Unless stated otherwise, state-averaged calculations were performed using the two lowest doublet states with equal weights. Mulliken spin densities of the state-averaged calculations were obtained from the first-order density matrix of the doublet ground state. The electronic energies of the different states can be found in Supplementary Data 3. Additional RASSCF calculations were performed by restricting the number of holes in all formally doubly occupied orbitals and the number of electrons in all formally virtual orbitals of the active space to two and four, respectively. A larger active space of 19 electron in 20 orbitals has been investigated at this level as well, including the bonding and anti-bonding orbitals of all $\sigma$-C-N bonds in addition to the previously mentioned orbitals.

Non-orthogonal CI (NOCI) calculations on DMP$^+$ were performed using the Q-Chem program package, Version 5.0[40], using the three qualitatively distinct UHF solutions discussed below as a quasidiabatic basis[41]. The UHF solutions were reliably generated using SCF metadynamics runs[42] on each structure. The employed penalty function had an initial width of 5.00 per electron and an initial height of 0.05 hartree. Upon convergence to one solution, orbitals were randomly mixed using a rotation angle of $\pi/8$.

Born-Oppenheimer ab initio molecular dynamics (AIMD) simulations on DMP$^+$ at the PBE0[43–45] DFT level were performed using the CP2K program package[46,47], DZVP-MOLOPT-GTH basis sets[48] combined with GTH-PBE pseudopotentials[49–51], and a D3 dispersion correction with Becke-Johnson damping[25,26,52]. For computing the electronic structure, we used the Quickstep (QS) code[47,53] employing the Gaussian Plane Wave (GPW) method[54] as implemented in CP2K. The SCF convergence criterion was set to $10^{-6}$. A cubic box of 25 Å length was used with periodic boundary conditions. After an initial structure optimization of the neutral DMP molecule, molecular dynamics simulations were carried out at 300 K, 565 K and 980 K controlled with a Nosé-Hoover chain thermostat[55,56] in a canonical ensemble (NVT). The MD timestep was chosen as 0.5 fs. After at least 5 ps of thorough equilibration, DMP was ionized to DMP$^+$ and a minimum of 5 trajectories of 5 ps length (each starting from different equilibration snapshots) were generated for analysis at each temperature. The initial and final structures of the trajectories are given in Supplementary Data 4.

### Calculations on DMP$^*$

Explicit structure optimizations for the Rydberg state of neutral DMP were performed using the linear-response spin-component scaled CC2 (LR-SCS-CC2) approach[57] implemented in the TURBOMOLE program, using aug-cc-pVTZ basis sets, and the appropriate auxiliary basis functions. The first excited state was generally targeted. A reaction path was again generated using the woelfling protocol (see above). The obtained structures can be found in Supplementary Data 2. Analogous optimizations of Rydberg-state stationary points were also performed at the LR-TDDFT level of theory[58,59] using the settings mentioned above. TDDFT calculations were performed using the GGA functional PBE[43,44], the global hybrids PBE0[45] and BHandHLYP[10] (all with DFT-D3

dispersion corrections, see above), and the range-separated hybrid $\omega$B97X-D[60]. Additional single-point energy calculations along the LR-SCS-CC2 reaction path were performed at the EOM-CCSD/aug-cc-pVTZ level, using the ORCA program (see above for further details).

### Reporting summary

Further information on research design is available in the Nature Portfolio Reporting Summary linked to this article.

## Data availability

The optimized molecular structures are provided in Supplementary Data 1 and 2. The total electronic energies generated in this study are provided in the Source Data file and in Supplementary Data 3. Initial and final structures of the molecular dynamics simulations can be found in Supplementary Data 4. Source data are provided with this paper. Additional data are available from the corresponding author upon request.

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

## Acknowledgements

The authors acknowledge funding by the German Science Foundation (Deutsche Forschungsgemeinschaft, DFG), project ID 387284271 (SFB 1349, M.R. and M.K.) and project ID 435886714 (Se1008/17-1, D.S. and C.K.).

## Author contributions

M.R., D.S. and M.K. conceived, D.S. and M.K. supervised the research. M.R. designed most of the methodology and performed most of the quantum chemical calculations. C.K. performed all MD simulations. M.R. and M.K. wrote the initial draft of the manuscript. All authors discussed the results and commented on the manuscript.

## Funding

## Competing interests

The authors declare no competing interests.
