## [Peer Review File · Nature Communications]

Rydberg Electron Stabilizes the Charge Localized State of the Diamine CationREVIEWER COMMENTS

Reviewer #1 (Remarks to the Author):

In this submission to Nature Communications, Kaupp and co-workers re-visit the interpretation of Rydberg spectra of gaseous dimethylpiperidine (DMP) as showing the co-existence of a localized and delocalized mixed-valent DMP⁺ radical cation. This topic has generated a controversial discussion about the performance of various electronic-structure methods, which was originally published 7 years ago in Nature Communications. The authors revisit this discussion by providing new high-level computations. The authors surprisingly and correctly find that the original calculations in the 2016 Nature Communications were carried out incorrectly and that follow up papers by the same authors to another journal (J. Phys. Chem. Lett. 2021, 12, 4, 1250–1255) also carried out calculations incorrectly. The authors attribute these errors due to unphysical curve crossings of the reference wave functions in the J. Phys. Chem. Lett. by the original authors. The authors then carry out new calculations to show that the phenomenon is actually a case where interactions of a Rydberg electron with the underlying cationic core alter molecular structure in such a fundamental way.

I find this manuscript to be of immense interest and brings closure to a highly contested issue that first started in the original 2016 Nature Communications paper. The authors correctly point out that the response by Wong and co-workers to the original paper (Nature Communications, 9, 4733 (2018)) was actually correct, and the cation is indeed a situation where DFT does not fail. As mentioned above, a follow-up paper in J. Phys. Chem. Lett. 2021, 12, 4, 1250–1255 claimed in its title to be a "Resolution of a Controversy" where they incorrectly claim that the response by Wong and co-workers is flawed. The authors of the current paper under review now find real closure on this issue to show that Wong and co-workers' response is actually correct and claims by Weber and co-workers were quite flawed and unfair to the response by Wong and co-workers. The authors clearly show this using their state-averaged multireference calculations as well as directly calculating the radical species, which shows agreement with experiment. As such, I strongly feel that this paper should be published in Nature Communications in its current form since these new findings clear up flawed arguments in the original 2016 Nature Communications paper by Weber and debunk their original claims that the cation cannot be described by DFT.

Reviewer #2 (Remarks to the Author):

The authors present a computational study of the two different problems. First, they revisit the potential energy curve of the DMP cation to re-investigate the heavily-argued case of the localized and delocalized states in DMP⁺. Second, they investigate the Rydberg states in the neutral DMP molecule and compare its electronic structure to the DMP ground state and the DMP cation. After carefully investigating this manuscript, I do not consider this work to be publishable in Nature Communications. My reasonings are listed below, divided into more general comments mentioning the flaws of the manuscript followed by the computational soundness of the presented study. In a nutshell, the majority of this work tries to reproduce the work of ref [arXiv:2007.06125], restricting the chosen methodology to conventional state-of-the-art computational methods without even considering actual high-level computational approaches, while their conclusions are not supported by the numerical data, which is not reproducible in parts. In short, there is no actual novelty in this work.

The authors claim that they provided sufficient proof to disregard the existence of the localized state of the DMP⁺ molecule. However, I disagree with their computational study, their discussion, and the interpretation of their results. Specifically,

1) the authors neglect important works in the literature. A high-level computational study has already been presented in ref [arXiv:2007.06125] three years ago. In the aforementioned work, large-scale DMRG calculations have been performed including a NEVPT2 correction. I do not see the point of doing such small CAS(11,12) calculations when better, actual high-level calculations have already been done.

2) the only originality of this work is the state-averaged calculations. But considering my first point, it is useless to perform such calculations on such a small active space (please see below for a more critical statement on their state-averaged methodology).

Their data analysis is inherently flawed. I will mention just a few of the major issues:

- state-averaged calculations are typically performed for states that are (almost) degenerate, very close in energy, or similar in electronic structure. I do not think that it is the case for DMP+. The numerical data presented in the manuscript does not confirm this either. Most importantly, the result of state-averaged calculations strongly depends on the choice of the weights used. Since the ground and first excited states are not degenerate, the choice of the weights will determine the final result. Obviously, one can play around with the weights and get the answer one wants.
- the crucial computations are not reproducible: the authors provide no information on the weights of their state-averaged CASSCF calculations, no information on the final orbitals, energies of each state, electronic wave functions, etc. These data are required to validate the quality of the calculations and to dissect the individual electronic structures. Only after such a careful analysis, any conclusions on the nature of the localized state can be made.
- the authors do not use actual highly-accurate state-of-the-art methodologies. Highly accurate calculations for such a tiny system as DMP+ can be performed with more advanced methods. Examples are DMRG (already done in arXiv:2007.06125), CCSDT or CCSDt, CR-EOM-CCSD(T), EOM-CCSDT, etc. To support their claims, I recommend using actual highly-accurate QC approaches, especially because they are computationally feasible for such a small system size (and small basis sets used)
- The errors in excitation energies of EOM-CCSD are around 0.25 eV (24 kJ/mol). Thus, the differences between the excitation energies and experimental data are within the error bound of the methods. If the authors wish to make any statements on the need for full triples, they have to perform the actual calculations.
- There are better methods than CCSD(T) to be called the gold standard of QC. Alternatives are the completely-renormalized version and flavors thereof. Furthermore, the authors (as also observed in previous works) highlight the significant differences in CCSD and CCSD(T) results. This points to the importance of triple excitations. Most likely, the perturbative treatment breaks down and an iterative triple correction has to be performed. I would not put too much interpretation in the CCSD(T) results without an actual analysis of the effect of T₃. Again, this is doable for this system.
- The authors represent their potential energy curve as a collection of equidistant points. However, they never specify how they constructed their potential energy surface and how this surface relates to the original work in ref 13 of the manuscript. Ref 13 contains roughly 9 points in each direction. However, the reaction path the authors chose most likely is not a straight line between two points. The authors should relate their reaction path to the original one and highlight the differences, especially considering their criticism against the original potential energy surface presented in ref 13.
- it appears that they report only vertical excitation energies for the Rydberg and ionized states. Geometry relaxation effects might be crucial here. Excited-state geometry optimization can be quickly done. It has been presented for various computational methodologies, even for actual highly-accurate methods like DMRG (see the works of Chan and others).

Based on the points mentioned above, I conclude that the presented computational methodology does not meet the standards in computational chemistry. The authors claim to use high-accurate methods, while they use the conventional toolbox that seems adequate for large-scale systems, but not for such a small model compound. Finally, the authors claim that they identified the DMP+ problem as an "artificial symmetry breaking and curve crossing." However, the numerical data does not support their conclusion (which is mostly based on state-averaged calculations) and appears to me as artificial symmetry restoration. The authors, need to significantly improve the computational study because I do not consider this work to be publishable even in a conventional quantum-chemistry-oriented journal.

Reviewer #3 (Remarks to the Author):

Rydberg Electron Stabilizes the Charge Localized State of the Diamine Cation

Reimann et al

This manuscript provides an alternative interpretation of a previous time-resolved photoelectron spectroscopic study on Rydberg excited states of dimethylpiperidine (DMP) in gas phase based on various high-level calculations. DMP is an interesting molecule which has inspired lots of experimental and theoretical studies, including a few controversial ones, because it shows the co-existence of a charge localized and delocalized mixed-valence DMP⁺ radical cation. By calculating the reaction paths for both the cationic states and the Rydberg states, the authors find that while a barrier exists in the Rydberg state, it vanishes in the cationic state, which is unusual since Rydberg electron is generally known for its weak binding and negligible interaction with the molecular ion core. In fact, the energy curves along the reaction coordinate between the delocalized and localized DMP structures show clearly that the curve of the Rydberg state is closely parallel to that of the ion state in the delocalized region (i.e., the Rydberg electron has negligible interaction with the molecular ion core, as known and common for Rydberg electrons) until the transition state and from there the curves have distinct trends and are no longer parallel in the localized region (i.e., the Rydberg electron has nonnegligible interaction with the molecular ion core and it does change the shape of the potential energy surface). The authors then further conclude that it is the strong binding of this Rydberg electron that stabilizes and generates a localized DMP* minimum, which does not exist for DMP⁺.

It is a very interesting finding that Rydberg electron plays such an important role in the structure determination. The calculations and interpretations seem to be sound and consistent with previous experiments. However, there are a few points the manuscript needs to address before acceptance:

1. The manuscript does not directly compare the calculations with the available observables (binding energies, relative energies of both the cations and the excited states, better to use consistent energy unit such as "eV" than a mixing use of "kJ/mol" and "eV"; more recently published ultrafast X-ray scattering signals on DMP) and evaluate the performance of various methods used in terms of agreement. The manuscript mentions a few energy comparisons in a way that is not easy to directly compare to the measurement.
2. Since a few methods are used and compared, it would be useful if the manuscript can identify the method or protocol that is suitable for such system and can be generalized.
3. The temperature for molecular dynamics calculation needs to be consistent with the experiment (much higher than 300K used in the calculation). It is worth noting that the localized and delocalized state's population depends heavily on the effective temperature in the system, as shown in ref. 8, different pump wavelengths will lead to different temperatures. Even the lowest temperature with ~240nm pump wavelength is significantly higher than 300K, however, there is almost no localized state signal at equilibrium. This may also explain why the localized state was not observed in solution, with even very polar solvent, as the solvent can serve as a cooling bath.
4. Continued with point 3, it is indeed important to compare apple to apple, as also showed in this manuscript. Therefore, a time-resolved direct measurement of gaseous DMP⁺ radical cation would make the conclusion much more convincing and add a lot of impact to the work.

Reviewer #1 (Remarks to the Author):

In this submission to Nature Communications, Kaupp and co-workers re-visit the interpretation of Rydberg spectra of gaseous dimethylpiperidine (DMP) as showing the co-existence of a localized and delocalized mixed-valent DMP⁺ radical cation. This topic has generated a controversial discussion about the performance of various electronic-structure methods, which was originally published 7 years ago in Nature Communications. The authors revisit this discussion by providing new high-level computations. The authors surprisingly and correctly find that the original calculations in the 2016 Nature Communications were carried out incorrectly and that follow up papers by the same authors to another journal (J. Phys. Chem. Lett. 2021, 12, 4, 1250–1255) also carried out calculations incorrectly. The authors attribute these errors due to unphysical curve crossings of the reference wave functions in the J. Phys. Chem. Lett. by the original authors. The authors then carry out new calculations to show that the phenomenon is actually a case where interactions of a Rydberg electron with the underlying cationic core alter molecular structure in such a fundamental way.

I find this manuscript to be of immense interest and brings closure to a highly contested issue that first started in the original 2016 Nature Communications paper. The authors correctly point out that the response by Wong and co-workers to the original paper (Nature Communications, 9, 4733 (2018)) was actually correct, and the cation is indeed a situation where DFT does not fail. As mentioned above, a follow-up paper in J. Phys. Chem. Lett. 2021, 12, 4, 1250–1255 claimed in its title to be a "Resolution of a Controversy" where they incorrectly claim that the response by Wong and co-workers is flawed. The authors of the current paper under review now find real closure on this issue to show that Wong and co-workers' response is actually correct and claims by Weber and co-workers were quite flawed and unfair to the response by Wong and co-workers. The authors clearly show this using their state-averaged multireference calculations as well as directly calculating the radical species, which shows agreement with experiment. As such, I strongly feel that this paper should be published in Nature Communications in its current form since these new findings clear up flawed arguments in the original 2016 Nature Communications paper by Weber and debunk their original claims that the cation cannot be described by DFT.

We thank the reviewer for this very positive assessment and agree with the notion that the present work does indeed provide the necessary "real" closure to the controversy and reconciles experiment and quantum chemistry for this question.

Reviewer #2 (Remarks to the Author):

The authors present a computational study of the two different problems. First, they revisit the potential energy curve of the DMP cation to re-investigate the heavily-argued case of the localized and delocalized states in DMP⁺. Second, they investigate the Rydberg states in the neutral DMP molecule and compare its electronic structure to the DMP ground state and the DMP cation. After carefully investigating this manuscript, I do not consider this work to be publishable in Nature Communications. My reasonings are listed below, divided into more general comments mentioning the flaws of the manuscript followed by the computational soundness of the presented study.

We disagree fundamentally with this assessment and will provide our counterarguments below.

In a nutshell, the majority of this work tries to reproduce the work of ref [arXiv:2007.06125], restricting the chosen methodology to conventional state-of-the-art computational methods without even considering actual high-level computational approaches, while their conclusions are not supported by the numerical data, which is not reproducible in parts. In short, there is no actual novelty in this work.

First of all, we object strongly to the notion that our work does not consider “actual high-level computational approaches”, which it indeed does. We note that both referees 1 and 3 even point this out explicitly (see also above). The basic misconception of reviewer 2, put forward also in refs. 8, 13, 14, is that the DMP⁺ cation is a multi-reference system and thus requires mandatorily multi-reference approaches with particularly large active spaces. This claim is contradicted clearly even by a basic chemical analysis of the system, as DMP exhibits a saturated σ -framework. A large active space necessarily includes the C-N and C-C σ -bonds in the active space (see below), which is completely unnecessary, as their contributions are excessively small in a final wave function of the radical cation and only add some dynamic and negligible static electron correlation. We had clearly shown already in our initial submission that single-reference methods like CCSD(T) are completely sufficient in this case, and the natural-orbital occupations point to a negligible entanglement of those σ -bonds to the hole(s) on the nitrogen atom(s). We now provide further evidence, including RASSCF calculations with large active spaces. We also reject the notion that our numerical data is not reproducible (see below) and find it peculiar that reviewer 2 argues specifically based on computational data in a non-peer-reviewed preprint.

The authors claim that they provided sufficient proof to disregard the existence of the localized state of the DMP⁺ molecule. However, I disagree with their computational study, their discussion, and the interpretation of their results. Specifically,

1) the authors neglect important works in the literature. A high-level computational study has already been presented in ref [arXiv:2007.06125] three years ago. In the aforementioned work, large-scale DMRG calculations have been performed including a NEVPT2 correction. I do not see the point of doing such small CAS(11,12) calculations when better, actual high-level calculations have already been done.

*See the above argument! Our new large-scale RASSCF calculations are completely consistent with our CASSCF and MRCI+Q data that show the state-specific calculations to exhibit artefacts derived from discontinuities. Moreover, we confirm in a multitude of ways that large-active-space multi-reference calculations are not needed to correctly describe the potential-energy-surface of DMP⁺. The reviewer does not seem to realize the symmetry-breaking problems of the state-specific calculations of ref. 14 (or of the preprint mentioned). In particular, when mentioning the DMRG+NEVPT2 calculations in that preprint, the referee apparently missed that adding dynamical correlation by NEVPT2 to the DMRG-CASSCF(19,20) wave function eliminates the barrier and thus the existence of DMP-L⁺! These calculations provide therefore no argument whatsoever **against** the soundness of our conclusions but rather **in their support**. Even though these data come from a non-peer-reviewed preprint, which is why we had not cited this work in our initial submission, we address these points now in a new section 2 in Supporting Information.*

2) the only originality of this work is the state-averaged calculations. But considering my first point, it is useless to perform such calculations on such a small active space (please see below for a more critical statement on their state-averaged methodology).

See our above arguments. It is not the state-averaged calculations that are irrelevant here but the use of large active spaces for a system that clearly does not constitute a multi-reference

case (see also comments by reviewer 1). While a larger active space helps bring in some dynamic correlation and thereby smoothen somewhat the area around the claimed barrier, it cannot compete with approaches like CCSD(T), which provide much more dynamical correlation and are clearly more appropriate for a case with only small static correlation. The latter point has now been bolstered additionally by a careful consideration of various multi-reference diagnostics in section 1 of Supporting Information. We also object to the incorrect statement that the state-averaged calculations are the only original part of our work (the most important part is in fact the role of the Rydberg electron!).

Their data analysis is inherently flawed. I will mention just a few of the major issues:
- state-averaged calculations are typically performed for states that are (almost) degenerate, very close in energy, or similar in electronic structure. I do not think that it is the case for DMP+. The numerical data presented in the manuscript does not confirm this either. Most importantly, the result of state-averaged calculations strongly depends on the choice of the weights used. Since the ground and first excited states are not degenerate, the choice of the weights will determine the final result. Obviously, one can play around with the weights and get the answer one wants.

This statement is incorrect and misses the context of the widespread use of state-averaging in the field to arrive at physically correct sets of orbitals. State-averaged multi-reference calculations are clearly not restricted to situations with near degeneracy but are also used, e.g., to avoid wave-function collapse for excited states, and so on. The state-averaging is necessary here because in the intermediate region along the reaction coordinate reference wave functions of two localized states become similarly important and generate an unphysical discontinuity in the MO composition when carrying out a state-specific calculation. What is claimed to be a barrier between a delocalized and localized minimum in refs. 8, 13, and 14 is in reality an artefact of the state-specific multi-reference calculations, but also of approaches with close relation to a UHF-type wave function, like their BHLYP calculations. These arise from a discontinuity of the underlying reference wave function. Reviewer 2 has apparently missed this fact. As the SA-CASSCF is used just to generate unbiased MOs along the reaction coordinate, equal weights are normally used for such purposes, and we followed this procedure to obtain physically correct orbitals along the entire reaction path. We show now in section 2 in Supporting Information that even a variation of weights between 4:1 and 1:1 does not affect these conclusions at all. We also underline the fact that our SA-CASSCF based curve very closely resembles the result of a non-orthogonal CI treatment based on the three different UHF solutions discussed in section 1 of the SI – which is a completely independent theoretical approach.

- the crucial computations are not reproducible: the authors provide no information on the weights of their state-averaged CASSCF calculations, no information on the final orbitals, energies of each state, electronic wave functions, etc. These data are required to validate the quality of the calculations and to dissect the individual electronic structures. Only after such a careful analysis, any conclusions on the nature of the localized state can be made.

The reviewer's statements are incorrect: while the weights had not been given, as they correspond to standard procedures (we state them now), we had provided the crucial Mulliken atomic spin densities at the relevant nitrogen atom to characterize the localization of the wave function and orbitals, and we had also already discussed the natural-orbital occupation numbers that reveal the absence of large static correlations. Further information can of course be used in any arbitrary combination but will not change the conclusions regarding the artificial symmetry breaking of the state-specific calculations and its correction by the state-averaged calculations.

- the authors do not use actual highly-accurate state-of-the-art methodologies. Highly accurate calculations for such a tiny system as DMP+ can be performed with more advanced methods. Examples are DMRG (already done in arXiv:2007.06125), CCSDT or CCSDt, CR-EOM-CCSD(T), EOM-CCSDT, etc. To support their claims, I recommend using actual highly-accurate QC approaches, especially because they are computationally feasible for such a small system size (and small basis sets used)

*In the non-peer-reviewed preprint emphasized by the reviewer, there is just a single calculation with a larger active space than the already large spaces we had used in our calculations. And as pointed out above, this active space is completely irrelevant given that DMP+ is not a multi-reference system. Moreover, we show now with RASSCF calculations (section 2 in Supporting Information) that the size of the active space does not change the underlying flaws of the state-specific calculations. For this system, DMRG cannot compete with CCSD(T), as it does not generate sufficient dynamical correlation. The addition of dynamical correlation by a DMRG+NEVPT2 approach eliminates the DMP-L+ minimum (as found in the preprint emphasized by the reviewer; see above). We are not aware of any published successful CCSDT, CCSDt, CR-EOM-CCSD(T) or EOM-CCSDT calculation on this system. In contrast to the opinion of the referee, these are by no means trivial calculations for the system at hand, at the limits of current computational capabilities, for most of the methods mentioned actually outside these limits. Our own attempts to run CCSDT calculations with programs like CFOUR or MRCC failed so far with meaningful basis-set sizes, due to the enormous core memory required and time limitations on the available compute cluster. Just before submission of this revision, Bryan Wong (UC Riverside), the corresponding author of ref. 12, informed me that they managed to finally run such CCSDT calculations and can confirm that they are in complete agreement with the CCSD(T) data. They plan to publish these results separately. This is just one more confirmation, in addition to all the overwhelming evidence presented in this work, that DMP+ is **not a multi-reference system**, and that **no DMP-L+ minimum exists!***

- The errors in excitation energies of EOM-CCSD are around 0.25 eV (24 kJ/mol). Thus, the differences between the excitation energies and experimental data are within the error bound of the methods. If the authors wish to make any statements on the need for full triples, they have to perform the actual calculations.

This statement is misleading. A deviation of 0.25 eV for an excitation energy of about 3 eV does not mean that the excited-state surface based on such calculations is inaccurate. Notably, we find very similar surfaces at various different, high levels of theory, such as (IP)-EOM-CCSD, LR-SCS-CC2 or various TDDFT levels.

- There are better methods than CCSD(T) to be called the gold standard of QC. Alternatives are the completely-renormalized version and flavors thereof. Furthermore, the authors (as also observed in previous works) highlight the significant differences in CCSD and CCSD(T) results. This points to the importance of triple excitations. Most likely, the perturbative treatment breaks down and an iterative triple correction has to be performed. I would not put too much interpretation in the CCSD(T) results without an actual analysis of the effect of T₃. Again, this is doable for this system.

These claims by the reviewer are not supported by any data and clearly contradicted by the results presented in our work. As we clearly show in Supporting Information, there is no indication that the perturbative (T) treatment should fail. The fact that there are indeed differences between CCSD and CCSD(T) simply points to the importance of triple excitations to correct the symmetry breaking of the UHF reference wave function. As pointed out also by

the authors of ref. 12 and referred to by reviewer 1, CCSD(T) is an entirely appropriate level for this cation, which is not a multi-reference system from any angle. For example, the magnitude of the D1 diagnostic, which has been discussed controversially in refs. 12, 13, simply indicates inaccuracies in the reference UHF wave function and does not give any indication that perturbational triple excitations are not appropriate. Indeed, the D1 diagnostic is minimal for the part of the surface, for which the perturbative triples contributions make the barrier vanish! Using Brueckner orbitals, for which the D1 diagnostic vanishes by construction, results in identical curves, signaling the inadequacy of this multi-reference diagnostic in this case. Other diagnostics such as the coefficient of the leading determinant in a CASSCF wave function, NOONs, or the %TAE(T) diagnostics, clearly exclude significant multi-reference character (section 2 in Supporting Information). These conclusions are further corroborated by our EOM-CCSD results based on orbitals of the closed-shell ground state: no barrier is found in this case, consistent with the very small barrier at UHF-CCSD level being an artefact of symmetry breaking. We emphasize all of the extensive evidence for the absence of multi-reference character!

- The authors represent their potential energy curve as a collection of equidistant points. However, they never specify how they constructed their potential energy surface and how this surface relates to the original work in ref 13 of the manuscript. Ref 13 contains roughly 9 points in each direction. However, the reaction path the authors chose most likely is not a straight line between two points. The authors should relate their reaction path to the original one and highlight the differences, especially considering their criticism against the original potential energy surface presented in ref 13.

We had clearly given the construction of the reaction path on p 5 of the manuscript and in the Methods section (now p. 12). The points are obtained from a quadratically optimized reaction path, they are not "equidistant points", as the reviewer mistakenly assumes! The text on p. 5 has been extended further for maximum clarity.

- it appears that they report only vertical excitation energies for the Rydberg and ionized states. Geometry relaxation effects might be crucial here. Excited-state geometry optimization can be quickly done. It has been presented for various computational methodologies, even for actual highly-accurate methods like DMRG (see the works of Chan and others).

We do not understand this statement, as our excited-state reaction coordinate is a fully relaxed LR-SCS-CC2-based one between fully optimized minima. We do not see where we could have missed any relevant relaxation effects.

Based on the points mentioned above, I conclude that the presented computational methodology does not meet the standards in computational chemistry. The authors claim to use high-accurate methods, while they use the conventional toolbox that seems adequate for large-scale systems, but not for such a small model compound. Finally, the authors claim that they identified the DMP+ problem as an "artificial symmetry breaking and curve crossing." However, the numerical data does not support their conclusion (which is mostly based on state-averaged calculations) and appears to me as artificial symmetry restoration. The authors, need to significantly improve the computational study because I do not consider this work to be publishable even in a conventional quantum-chemistry-oriented journal.

We have clearly shown above why these statements by reviewer 2 are incorrect. We emphasize that reviewer 1 agrees with our notion that the conclusions of ref. 14 (and by inference those in the preprint reviewer 2 refers to) are erroneous, and so are the statements of reviewer 2. The reviewer ignores the conclusive observation that DMP⁺ is clearly not a

multi-reference system. He/she also does not seem to acknowledge that we showed the state-specific CASSCF and MRCI calculations of ref. 14 in a definite manner to be erroneous due to discontinuities. The “symmetry restoration” by state-averaging is by no means artificial but the correct and accepted procedure in such cases, and we show that our results are consistent with many additional independent methods such as EOM-CCSD and NOCI based on three different UHF solutions. These points clearly invalidate the criticism of the reviewer. We also object to the notion that we have used anything but the appropriate state-of-the-art computational methodology at the highest available levels.

Reviewer #3 (Remarks to the Author):

This manuscript provides an alternative interpretation of a previous time-resolved photoelectron spectroscopic study on Rydberg excited states of dimethylpiperidine (DMP) in gas phase based on various high-level calculations. DMP is an interesting molecule which has inspired lots of experimental and theoretical studies, including a few controversial ones, because it shows the co-existence of a charge localized and delocalized mixed-valence DMP⁺ radical cation. By calculating the reaction paths for both the cationic states and the Rydberg states, the authors find that while a barrier exists in the Rydberg state, it vanishes in the cationic state, which is unusual since Rydberg electron is generally known for its weak binding and negligible interaction with the molecular ion core. In fact, the energy curves along the reaction coordinate between the delocalized and localized DMP structures show clearly that the curve of the Rydberg state is closely parallel to that of the ion state in the delocalized region (i.e., the Rydberg electron has negligible interaction with the molecular ion core, as known and common for Rydberg electrons) until the transition state and from there the curves have distinct trends and are no longer parallel in the localized region (i.e., the Rydberg electron has nonnegligible interaction with the molecular ion core and it does change the shape of the potential energy surface). The authors then further conclude that it is the strong binding of this Rydberg electron that stabilizes and generates a localized DMP* minimum, which does not exist for DMP⁺.

It is a very interesting finding that Rydberg electron plays such an important role in the structure determination. The calculations and interpretations seem to be sound and consistent with previous experiments. However, there are a few points the manuscript needs to address before acceptance:

We thank the reviewer for the overall very positive assessment of both the quality of the computations and the importance of our findings.

1. The manuscript does not directly compare the calculations with the available observables (binding energies, relative energies of both the cations and the excited states, better to use consistent energy unit such as “eV” than a mixing use of “kJ/mol” and “eV”; more recently published ultrafast X-ray scattering signals on DMP) and evaluate the performance of various methods used in terms of agreement. The manuscript mentions a few energy comparisons in a way that is not easy to directly compare to the measurement.

We have now tried to make the comparison with the experimental data clearer and use exclusively eV as an energy unit throughout. We do now provide additionally explicit binding energies at various levels for a direct comparison with experimental data in Supporting Information (Table S2). We also include a discussion of the available structural parameters obtained from X-ray scattering on p. 9 of the main text and an additional Table S3 in Supporting Information. We are grateful to the reviewer for pointing out the recent X-ray scattering study (now cited as ref. 9), which had escaped our literature search.

2. Since a few methods are used and compared, it would be useful if the manuscript can identify the method or protocol that is suitable for such system and can be generalized.

We do now point on p. 10 of the main text to LR-SCS-CC2 and (IP)-EOM-CCSD as particularly appropriate methods to treat the Rydberg-state surface, even though various TDDFT approaches also seem to provide accurate results. We also point out explicitly (e.g. on the bottom of p. 6) that CCSD(T) is a perfectly appropriate high-level method to treat the fully ionized surface, due to the absence of strong multi-reference character. Again, various other approaches, including many DFT functionals, provide correct results as well for the cation.

3. The temperature for molecular dynamics calculation needs to be consistent with the experiment (much higher than 300K used in the calculation). It is worth noting that the localized and delocalized state's population depends heavily on the effective temperature in the system, as shown in ref. 8, different pump wavelengths will lead to different temperatures. Even the lowest temperature with ~240nm pump wavelength is significantly higher than 300K, however, there is almost no localized state signal at equilibrium. This may also explain why the localized state was not observed in solution, with even very polar solvent, as the solvent can serve as a cooling bath.

We thank the reviewer for pointing this out! We have extended the AIMD simulations and cover now also trajectories at 565 K and at 980 K, to be as close to the experimental setup as possible. These are the lowest and highest temperatures, respectively, that would correspond to the total internal energy deposited in the pump-probe experiments. The amplitudes of the motions increase of course for the higher temperatures, in particular at 980 K, and the system therefore samples a bit more the region of dihedral angles corresponding to a DMP-L⁺-like structure. However, at no temperature do we find a situation where an extra localized minimum on the free-energy surface would be realistic. Our original conclusions thus remain unchanged.

4. Continued with point 3, it is indeed important to compare apple to apple, as also showed in this manuscript. Therefore, a time-resolved direct measurement of gaseous DMP⁺ radical cation would make the conclusion much more convincing and add a lot of impact to the work.

A time-resolved spectroscopic study of DMP⁺ in the gas phase would indeed be desirable but is clearly outside the scope of our work. We suspect that this may be a formidable challenge given the rarity of such studies in general, in particular in a time-resolved fashion. Clearly this must have been one of the reasons why the authors of refs. 7, 8 chose to look instead at the Rydberg state of neutral DMP.

REVIEWER COMMENTS

Reviewer #1 (Remarks to the Author):

I fully support the publication of this manuscript in its current form. I was already extremely supportive of the first iteration of this manuscript since it brings closure to a highly contested issue that first started in the original 2016 Nature Communications paper. As mentioned in my first review, the authors of the current paper under review find real closure on this issue to show that Wong and co-workers' response is actually correct and claims by Weber and co-workers were quite flawed.

I have also fully assessed Reviewer 2's comments, which I find not to be factual. In particular, reviewer 2 mentions an ArXiv paper (arXiv:2007.06125) as being a "high-level computational study that has already been presented three years ago." However, this ArXiv paper is just a pre-print of a now-published paper (J. Phys. Chem. Lett. 2021, 12, 4, 1250–1255), which Kaupp and co-workers are showing is incorrect. In other words, the ArXiv paper and the J. Phys. Chem. Lett. paper is the very same work, and Reviewer 2 is intentionally trying to confuse this issue (possibly to prevent embarrassment to the authors of the ArXiv/JPCL papers with very incorrect results).

Again, this paper should be published as soon as possible to bring closure and clarify this issue to the scientific community on this hotly debated issue.

Reviewer #2 (Remarks to the Author):

In the revised version of their manuscript, the authors have included some additional data to show the „correctness“ of their initial state-average calculations. Specifically, they performed state-average calculations with different weights and included new RAS-SCF calculations to approximate the DMRG results. The results of these additional calculations are rather obvious, and the final RAS-SCF results approach the DMRG reference results from a previous work (published on arXiv). Even though I appreciate the small amount of effort the authors spent in producing this additional numerical data, I do not consider their responses to my comments appropriate and sufficient. As they claim themselves in their SI, [„The DMRG+NEVPT2 results are fully consistent with the conclusions of the present work regarding the absence of a DMP-L+ minimum, and with those of Ref. 4 regarding the accuracy of single-reference CCSD(T)“], one part of their computational study is a reproduction of existing results (peer-reviewed or arXiv). Thus, I am missing any new physical insights regarding the DMP-L+ minimum. Furthermore, the authors claim in their rebuttal letter that the „most important part is in fact the role of the Rydberg electron,“ but in the manuscript, they state that [„In this work we resolve the controversial discussion of the simultaneous existence of a delocalized and localized mixed-valent (MV) minimum of the prototypical DMP+ cation.“] The authors cannot have it both ways.

In my opinion, this manuscript deals with a computational difficulty, which indeed can be solved (according to the claims of the authors) using state-of-the-art methods or has already been shown by others (also confirmed by the authors). Therefore, the degree of new physical insight does not justify a publication in Nature Communications, but in a more computationally oriented journal with a more suitable readership (like Journal of Physical Chemistry Letters or Journal of Chemical Theory and Computation) or, alternatively, a transfer to Scientific Reports if my comments below are adequately addressed.

- 1) The authors consistently claim that the DMP+ molecule is not a multi-reference system. This raises the question of why the authors performed CAS-SCF calculations in the first place. Why perform state-average CAS-SCF or RAS-SCF if single-reference methods are sufficient?
- 2) The analysis of whether to state-average or not addresses a very technical question, not suitable for the readership of Nature Communications.

- 3) Please provide a reference which states explicitly that equal weights in state-average calculations are the way how these calculations should be performed. Section 2, Figures 8 and 9 in the SI indeed confirm my original comment: depending on the choice of the weights, one can „steer“ the outcome of the results. The outcome of Figure 9 is obvious and what you would expect from changing the weights in state-average calculations, while Figure 8 demonstrates how CAS-SCF and RAS-SCF approach the DMRG reference results when increasing the active space. This is no new physical insight, just a simple confirmation of expected behavior and how to approach the DMRG reference results.
- 5) According to their new „computational details“, the state-average calculations have been performed for the two lowest singlet states. DMP+ has a doublet ground states. Please clarify.
- 6) Why are the authors so hesitant in presenting their numerical results to allow for reproducibility of their results and assessment of the soundness of their calculations?
- CAS-SCF/RAS-SCF results must be presented with the final (natural) orbitals. Please note that simply stating the choice of active space orbitals does not make the computations reproducible.
 - For the state-average case, give the corresponding natural orbitals for the averaged state and compare them to the state-specific ones. This allows for a direct comparisons of the underlying wave functions.
 - Provide the energies of all CAS-SCF, RAS-SCF, and state-averaged states. How large is the difference between the ground and the first excited state?
 - Considering the reaction pathway: How was each point chosen (that is, what coordinates have been modified), or what is the rationale for designing this new path Provide a figure where the new pathway is compared to the original one of Ref. 14. If not possible, provide a figure of the corresponding molecular structures in the SI. The choice of the reaction coordinate is still unclear.
 - Please, scrutinize why this new reaction pathway is better than the original one by stating more than just the obvious „more number of points“ explanation.
- 7) The figures in the manuscript (and SI) are horrible. There is too much information in one figure. In Figure 1, separate the left and right y-axis and make two plots. Include proper legends.
- 8) In their rebuttal letter, the authors argue in terms of correlation and entanglement. However, they never „measure“ any correlation or entanglement encoded in each quantum state. They argue in terms of occupation numbers or diagnostic tools. Note that these measures do not quantify any correlations or entanglement in electronic wave functions. These measures are used to validate a computational methodology. If the authors wish to make statements on actual correlation and entanglement, they need to use appropriate measures. Otherwise, I recommend avoiding such statements.
- 9) DMRG is more than just a multi-reference method that allows us to study large active spaces. The power of DMRG is that it allows for a balanced description of static and dynamic electron correlation. CAS-SCF typically overestimates the former, while DMRG cannot fully account for the latter. It has been shown in the literature that DMRG provides a proper description of static correlation by including parts of the dynamic correlation. Since DMRG does not account for dynamic correlation outside the active space, corrections must be added. However, this task is not trivial, and, in my opinion, currently, no satisfying a posteriori correction exists (possibilities are NEVPT2, CASPT2, tailored CC, etc.). Moreover, NEVPT2 results have to be considered with care. A NEVPT2 correction does not always improve in the right direction. It is known in the literature that NEVPT2 can result in larger errors than CASPT2. My point here is that CAS-SCF might have the same problem in terms of electron correlation, which DMRG might be able to correct (at least for the localized state). I am not talking about energies but correlation and entanglement. In my opinion, this manuscript does not go beyond electronic energies. I do not consider the Mulliken SDs to give a qualitative measure for correlation or entanglement. If the authors insist in presented their CAS-SCF or RAS-SCF results in the paper, they need to provide additional insights and discussion (beyond energies). Their numerical study on CAS-SCF and RAS-SCF reproduces (or better approaches) literature results. They must go beyond that. State-averaging will not solve or answer for the discrepancies between CAS-SCF and DMRG (or RAS-SCF, as it seems to be similar to DMRG here).
- 10) If the authors disagree and that DMRG is not sufficient and CAS-SCF (state-specific and state-average) is sufficient, they must provide numerical evidence to support their claims. That means, DMRG data that indeed indicates that. Currently, their RAS-SCF results point to a different conclusion.
- 11) If state-averaging is important, the authors must perform state-average DMRG calculations. By doing so, electron correlation is properly described in the active space (CAS-SCF is not good

enough) and they will be able to average over more states. DMRG is available for free in many QC codes, like Orca or OpenMolcas. PySCF should also be able to provide an interface.

Reviewer #3 (Remarks to the Author):

Rydberg Electron Stabilizes the Charge Localized State of the Diamine Cation
Revised version A

Reimann et al

This revised manuscript has added a few details to make the main texts and supporting information clearer. The authors have also responded to the referees' comments point-by-point. While some of them are adequate, some of them are still unsatisfactory. In particular, the response "A time-resolved spectroscopic study of DMP+ in the gas phase would indeed be desirable but is clearly outside the scope of our work. We suspect that this may be a formidable challenge given the rarity of such studies in general, in particular in a time-resolved fashion. Clearly this must have been one of the reasons why the authors of refs. 7, 8 chose to look instead at the Rydberg state of neutral DMP." to my previous comment is not satisfactory in a few ways. Firstly and most importantly, the request for more and stronger experimental evidence of DMP+ is clearly within the scope because the manuscript claims to discover an unusual (and the first) case where Rydberg electron interacts with the cationic core so much that it alters the molecular structure in a fundamental way, which naturally requires a comparison between the Rydberg state and the cation. The fact that there are two interpretations with opposite opinions about the multi-reference character of the DMP+ system, which both seem to be able to explain the existing experimental observables in the references, emphasizes the importance of the direct measurement of gas-phase DMP+ cation. "This work presents the first case where interactions of a Rydberg electron with the underlying cationic core alter molecular structure in such a fundamental way" is a big claim if it is true, which definitely requires experimental support. Secondly, the authors of the manuscript "suspect that the direct measurement of gas-phase DMP+ may be a formidable challenge" and "one reason the authors of refs. 7, 8 chose to look instead at the Rydberg state of neutral DMP". This is not convincing to me. The Rydberg electron is widely and long-time known to have small interaction with thus negligible effect on the cationic core, so it is not surprising people would assume similar structures of them and use the Rydberg state to study the cation, as the Rydberg electron is easy to detach and used as a probe. While I agree that Rydberg state is more convenient to be used to study cation when it is appropriate (as in most cases), it is not formidable challenge to study the cation directly and should be carried out in order to prove the rarer cases when the previously mentioned scenario fails (as claimed in this manuscript, and why the manuscript is interesting in the first place). The gas-phase neutral DMP can be readily ionized by one VUV (~147nm) photon or two UV (~294nm) photons to generate DMP+ and probed by various methods such as structure-sensitive X-ray or electron scattering, or state-sensitive white light, or site-sensitive X-ray absorption or photoelectron spectroscopy. Without the necessary experimental support, the manuscript is an interesting alternative explanation of previous data, but not more.

In addition, the connections to the previous experiments are still inadequate, or sometimes confusing. For example, even though the authors responded, "We do now point on p. 10 of the main text to LR-SCS-CC2 and (IP)-EOM-CCSD as particularly appropriate methods to treat the Rydberg-state surface, even though various TDDFT approaches also seem to provide accurate results.", there are no direct comparison and discussion about the agreements with the experimental binding energies and relative energies for LR-SCS-CC2 and TDDFT. There is a short discussion of the EOM-CCSD results, but as this has been reported in reference 8 and the arXiv preprint reviewer 2 mentioned. In particular, according to the Supplementary Table 2, LR-SCS-CC2 underestimates the binding energy of the DMP-D* state and overestimates the binding energy of

the DMP-L* state. It also overestimates the binding energy difference a lot. Most of the TDDFT results listed in the table are far from "accurate" comparing to the experimental data. On the other hand, the PZ-SIC results reported in reference 8 match the experiment much better. It is difficult to justify the choice of method when the method does not match the experimental data as well as a previously published work.

Another two examples of confusing connections to the previous experiments are both on page 10. One is "The change in binding energies of the Rydberg electron also matches the observed experimental increase of about 0.11 eV from the localized to the delocalized state.⁷" Apparently, the binding energy of the delocalized state is lower, not higher, than that of the localized state according to the cited reference 7. The other one is "The LR-SCS-CC2/aug-cc-pVTZ energy of DMP-L* relative to DMP-D* (0.24 eV) underestimates somewhat the experimental estimate of 0.33 ± 0.04 eV obtained from thermal populations.⁸" The 0.33 ± 0.04 eV in the cited reference 8 is the energy difference between the cations after considering the binding energy difference. There is one more confusing connection on page 12 of the supporting information. "but even at 980K structures similar to DMP-L+ a) do not account for a significant maximum in the combined distribution functions and b) are not observed immediately following ionization as suggested by experiments.³" How is this suggested by the experiments in the SI reference 3? The experiments apparently observed a localized state in the beginning at and shortly after t_0 .

In addition, the respond "This claim is contradicted clearly even by a basic chemical analysis of the system, as DMP exhibits a saturated sigma-framework. A large active space necessarily includes the C-N and C-C sigma-bonds in the active space (see below), which is completely unnecessary, as their contributions are excessively small in a final wave function of the radical cation and only add some dynamic and negligible static electron correlation." does not consider that the C-C sigma* anti-bonds participate in the mixing with the nitrogen lone pairs to assist the charge delocalization procedure via the so called "through-bond-interaction", as supported by the weakened elongated C-C bond in the X-ray scattering experiment in reference 9.

There are also a few minor items that are needed to be improved. Firstly, some of the figures are hard to read. For example, Figure 1 has many curves in one plot, it would be easier to follow which is which with a proper legend of the curves and dots. The same suggestion goes to the figures in the SI too when there are many curves and dots. One can improve the readability of figures by changing the type of the curve and the shape of the plot symbols and using proper legend. Secondly, it would be clearer to mark which part or side of the plots is localized and which is delocalized. Some of the figures have the label but many do not. Thirdly, the real experiment is from the localized state to the delocalized state, it is a little counter-intuitive to have the reaction path reversed.

In summary, this manuscript has interesting ideas that may change our way of thinking of Rydberg electrons and their effects on the molecular cationic cores. However, it lacks the critical experimental data to support these ideas. These data must be obtained before the conclusion can be drawn.

Reviewer #1 (Remarks to the Author):

I fully support the publication of this manuscript in its current form. I was already extremely supportive of the first iteration of this manuscript since it brings closure to a highly contested issue that first started in the original 2016 Nature Communications paper. As mentioned in my first review, the authors of the current paper under review find real closure on this issue to show that Wong and co-workers' response is actually correct and claims by Weber and co-workers were quite flawed.

I have also fully assessed Reviewer 2's comments, which I find not to be factual. In particular, reviewer 2 mentions an ArXiv paper (arXiv:2007.06125) as being a "high-level computational study that has already been presented three years ago." However, this ArXiv paper is just a pre-print of a now-published paper (J. Phys. Chem. Lett. 2021, 12, 4, 1250–1255), which Kaupp and co-workers are showing is incorrect. In other words, the ArXiv paper and the J. Phys. Chem. Lett. paper is the very same work, and Reviewer 2 is intentionally trying to confuse this issue (possibly to prevent embarrassment to the authors of the ArXiv/JPCL papers with very incorrect results).

Again, this paper should be published as soon as possible to bring closure and clarify this issue to the scientific community on this hotly debated issue.

We thank the reviewer for the positive assessment.

Reviewer #2 (Remarks to the Author):

In the revised version of their manuscript, the authors have included some additional data to show the „correctness” of their initial state-average calculations. Specifically, they performed state-average calculations with different weights and included new RAS-SCF calculations to approximate the DMRG results. The results of these additional calculations are rather obvious, and the final RAS-SCF results approach the DMRG reference results from a previous work (published on arXiv). Even though I appreciate the small amount of effort the authors spent in producing this additional numerical data, I do not consider their responses to my comments appropriate and sufficient. As they claim themselves in their SI, [„The DMRG+NEVPT2 results are fully consistent with the conclusions of the present work regarding the absence of a DMP-L+ minimum, and with those of Ref. 4 regarding the accuracy of single-reference CCSD(T)”, one part of their computational study is a reproduction of existing results (peer-reviewed or arXiv). Thus, I am missing any new physical insights regarding the DMP-L+ minimum. Furthermore, the authors claim in their rebuttal letter that the „most important part is in fact the role of the Rydberg electron,” but in the manuscript, they state that [„In this work we resolve the controversial discussion of the simultaneous existence of a delocalized and localized mixed-valent (MV) minimum of the prototypical DMP+ cation.”] The authors cannot have it both ways.

All of these statements are plainly wrong. The DMRG results from a preprint, that the referee refers to over and over again, are by no means reference results, for reasons we will further detail (again!) below. Therefore, our multi-reference calculations have never been about reproducing these results but to show that they are flawed, and that their interpretation, and that of the MRCI calculations from ref. 14, is erroneous and leads to the misconception at the heart of the controversy. The statement “The authors cannot have it both ways.” is simply nonsense. We have clearly resolved the discussion on the cation by showing that the state-

specific MRCI and some other calculations erroneously produced a barrier, and we then moved on and showed that the influence of the Rydberg electron is needed to reproduce the experimental observations. If the reviewer cannot understand these facts, he/she is unsuitable to review this work. The statement "I am missing any new physical insights regarding the DMP-L⁺ minimum" proves this!

In my opinion, this manuscript deals with a computational difficulty, which indeed can be solved (according to the claims of the authors) using state-of-the-art methods or has already been shown by others (also confirmed by the authors). Therefore, the degree of new physical insight does not justify a publication in Nature Communications, but in a more computationally oriented journal with a more suitable readership (like Journal of Physical Chemistry Letters or Journal of Chemical Theory and Computation) or, alternatively, a transfer to Scientific Reports if my comments below are adequately addressed.

See previous comment. The completely novel physical insight, acknowledged clearly by reviewers #1 and #3, is that a Rydberg electron can have such a profound influence on the structural and dynamical properties of a cationic core. Neglecting this shows that the reviewer is neither serious nor impartial.

1) The authors consistently claim that the DMP⁺ molecule is not a multi-reference system. This raises the question of why the authors performed CAS-SCF calculations in the first place. Why perform state-average CAS-SCF or RAS-SCF if single-reference methods are sufficient?

DMP⁺ is indeed no multi-reference system. As pointed out above and in our previous rebuttal, over and over again, we performed multi-reference calculations to (successfully) demonstrate the flaws in the previous MRCI results on the cation. We do not think that this should be hard to understand.

2) The analysis of whether to state-average or not addresses a very technical question, not suitable for the readership of Nature Communications.

This analysis of the state-averaging is clearly required to show the errors in the previous interpretation of the experimental observations via the isolated cation. We do already have most of the details in the Supporting Information, but the core arguments are clearly needed in the main text to make this important point. The referee again obfuscates matters.

3) Please provide a reference which states explicitly that equal weights in state-average calculations are the way how these calculations should be performed.

While in our experience this is indeed typical practice in the field for excited-state calculations, there is no particular work where this is stated as a rule. It is, however, generally common to adjust the weights to obtain a balanced description of the orbitals (see, e.g., DOI: 10.1016/j.jms.2015.02.016), as we have done in this work.

Section 2, Figures 8 and 9 in the SI indeed confirm my original comment: depending on the choice of the weights, one can „steer“ the outcome of the results.

Of course, the weights influence the results. However, Supplementary Figure 9 clearly demonstrates that the MRCI+Q results are much less sensitive to the weights than the

CASSCF reference, and they can eliminate the unphysical barrier (kink) of the latter. Throughout this report and the previous one, the referee fails to acknowledge this unphysical kink and its consequences and seems to avoid addressing this matter.

The outcome of Figure 9 is obvious and what you would expect from changing the weights in state-average calculations, while Figure 8 demonstrates how CAS-SCF and RAS-SCF approach the DMRG reference results when increasing the active space. This is no new physical insight, just a simple confirmation of expected behavior and how to approach the DMRG reference results.

As stated further above, the DMRG results can by no means be viewed as reference results (see also below), and Supplementary Figure 8 serves merely to show that RASSCF and DMRG with the same active spaces give a very similar answer (which is not the final and correct answer!).

5) According to their new „computational details”, the state-average calculations have been performed for the two lowest singlet states. DMP+ has a doublet ground states. Please clarify.

This typo was corrected on pp. 6 and 13.

6) Why are the authors so hesitant in presenting their numerical results to allow for reproducibility of their results and assessment of the soundness of their calculations?
- CAS-SCF/RAS-SCF results must be presented with the final (natural) orbitals. Please note that simply stating the choice of active space orbitals does not make the computations reproducible.

To remove even the smallest (in our opinion unreasonable) doubt that our multi-reference calculations are reproducible, we have now added pictures of the final natural orbitals to the Supporting Information on p. 16 (section 6 of the SI).

- For the state-average case, give the corresponding natural orbitals for the averaged state and compare them to the state-specific ones. This allows for a direct comparisons of the underlying wave functions.

We have added pictures of the final natural orbitals to the Supporting Information on p. 17 (section 6 of the SI).

- Provide the energies of all CAS-SCF, RAS-SCF, and state-averaged states. How large is the difference between the ground and the first excited state?

We have added the electronic energies of all calculations to the Supporting Information. The energy differences can be obtained easily from these numbers.

- Considering the reaction pathway: How was each point chosen (that is, what coordinates have been modified), or what is the rationale for designing this new path Provide a figure where the new pathway is compared to the original one of Ref. 14. If not possible, provide a figure of the corresponding molecular structures in the SI. The choice of the reaction coordinate is still unclear.

The construction is clearly described on p. 12. It is the direct minimum energy path (we have now added the words “minimum energy” to avoid any misunderstanding) on the potential energy surface at BHandHLYP level for the cation and at LR-SCS-CC2 level for the Rydberg state, and it does not depend trivially on any one coordinate. All structures of the two paths had already been given in the SI so that the path can be easily reproduced (and a mapping to any other desired path can be performed).

- Please, scrutinize why this new reaction pathway is better than the original one by stating more than just the obvious „more number of points” explanation.

We had already clearly stated on p. 5 that this new path is free from the dependence on any particular internal coordinate and thus of any potentially biased choice of coordinates by the scientist carrying out the calculations.

7) The figures in the manuscript (and SI) are horrible. There is too much information in one figure. In Figure 1, separate the left and right y-axis and make two plots. Include proper legends.

While we strongly object to the description as “horrible”, we have expanded the legends (see also below). Splitting the plot into two plots makes no sense, however, as this would not allow the correlation between the energy cusp and the switching of the spin densities to be seen.

8) In their rebuttal letter, the authors argue in terms of correlation and entanglement. However, they never „measure” any correlation or entanglement encoded in each quantum state. They argue in terms of occupation numbers or diagnostic tools. Note that these measures do not quantify any correlations or entanglement in electronic wave functions. These measures are used to validate a computational methodology. If the authors wish to make statements on actual correlation and entanglement, they need to use appropriate measures. Otherwise, I recommend avoiding such statements.

While we had mentioned entanglement in the previous rebuttal letter, it is nowhere discussed in the manuscript. The NO occupations help understand the degree of multi-reference character (actually its absence) and are thus indirect means of demonstrating the lack of entanglement. To open a lengthy discussion on entanglement in the paper would be counterproductive for the reader; this seems to be another deliberate attempt by the reviewer to obfuscate matters.

9) DMRG is more than just a multi-reference method that allows us to study large active spaces. The power of DMRG is that it allows for a balanced description of static and dynamic electron correlation. CAS-SCF typically overestimates the former, while DMRG cannot fully account for the latter. It has been shown in the literature that DMRG provides a proper description of static correlation by including parts of the dynamic correlation. Since DMRG does not account for dynamic correlation outside the active space, corrections must be added.

This entire statement is completely nonsensical. DMRG uses tensor-product approximations to efficiently approximate a CASSCF or full-CI with the same active space, and it thereby allows larger active spaces. Nothing more and nothing less! The statement that DMRG allows for a balanced description of static and dynamical correlation is contradicted immediately in the next sentence of the reviewer, where he/she admits that DMRG does not fully cover dynamical correlation. To be precise, DMRG would cover both if used with all

orbitals (including all virtual ones) in the active space and if going to the complete basis-set limit. This is not at all the case in the cited preprint, and in fact the basis set used is very far from convergence to cover dynamical correlation (large angular momentum functions are required to fulfill the interelectronic cusp condition). Therefore, any currently possible DMRG calculations on a system like DMP^+ are just approximations to CASSCF with relatively large active spaces, and they cover only a very small fraction of dynamical correlation. We have now added a statement on this matter to the main text (pp. 6-7). The claim that DMRG constitutes a reference calculation in the present context is thus utter nonsense. The statement “DMRG provides a proper description of static correlation by including parts of dynamical correlation.” also is a complete contradiction in itself. Any CCSD or CCSD(T) calculation covers much more dynamical correlation, and even a good DFT calculation does. This is now shown explicitly for CCSD on p. 9 in Supplementary Information (note that deviations of CCSD from variationality are much smaller than the energy differences discussed).

However, this task is not trivial, and, in my opinion, currently, no satisfying a posteriori correction exists (possibilities are NEVPT2, CASPT2, tailored CC, etc.). Moreover, NEVPT2 results have to be considered with care. A NEVPT2 correction does not always improve in the right direction. It is known in the literature that NEVPT2 can result in larger errors than CASPT2. My point here is that CAS-SCF might have the same problem in terms of electron correlation, which DMRG might be able to correct (at least for the localized state).

This is another puzzling contradiction by the reviewer of his/her own statements: if DMRG is a reference method, why would it need corrections? In the present context, DMRG+NEVPT2 can be expected to give a reasonable result: in cases of small static correlation, or if DMRG adequately covers static correlation, the PT2 part only needs to cover dynamical correlation, and it does so comparably well as in single-reference MP2. Note also that DMRG cannot be better than CASSCF in the same active space, another patently false statement by the reviewer!

I am not talking about energies but correlation and entanglement. In my opinion, this manuscript does not go beyond electronic energies. I do not consider the Mulliken SDs to give a qualitative measure for correlation or entanglement.

As this paper focusses on energies and does not discuss entanglement at all (see above), this is another completely irrelevant point. The Mulliken SDs characterize the localization of charge, however, which is important in the context of this work.

If the authors insist in presented their CAS-SCF or RAS-SCF results in the paper, they need to provide additional insights and discussion (beyond energies). Their numerical study on CAS-SCF and RAS-SCF reproduces (or better approaches) literature results. They must go beyond that. State-averaging will not solve or answer for the discrepancies between CAS-SCF and DMRG (or RAS-SCF, as it seems to be similar to DMRG here).

As discussed above, the CASSCF and RASSCF calculations are provided only in order to show that the previous CASSCF and MRCI calculations were flawed, a point the reviewer again deliberately tries to obfuscate! And doing the state averaging is exactly what is needed to go beyond the previous calculations and to rectify their shortcomings (their unphysical discontinuity on the potential energy surface at CASSCF level). In fact, the CASSCF and RASSCF calculations, which indeed provide the necessary insights and clearly go beyond the

previous work, are anyway given only on pp. 9-11 of the SI. Therefore, this point is just more unnecessary smoke by the reviewer.

10) If the authors disagree and that DMRG is not sufficient and CAS-SCF (state-specific and state-average) is sufficient, they must provide numerical evidence to support their claims. That means, DMRG data that indeed indicates that. Currently, their RAS-SCF results point to a different conclusion.

This is again a completely nonsensical statement: we did not claim that CASSCF is notably better than DMRG, when the same active space and basis-sets are used. Both lack most of dynamical correlation (we have added a corresponding statement on pp. 6-7). CASSCF is used by us as a starting point for MRCI+Q and not as a benchmark level at all.

11) If state-averaging is important, the authors must perform state-average DMRG calculations. By doing so, electron correlation is properly described in the active space (CAS-SCF is not good enough) and they will be able to average over more states. DMRG is available for free in many QC codes, like Orca or OpenMolcas. PySCF should also be able to provide an interface.

And just another completely nonsensical statement by the reviewer! As stated above, DMRG approximates CASSCF in the same active space, why should it be better than the latter? Any consequences of state averaging at CASSCF level will hold also at the corresponding DMRG level.

Reviewer #3 (Remarks to the Author):

Rydberg Electron Stabilizes the Charge Localized State of the Diamine Cation
Revised version A

Reimann et al

This revised manuscript has added a few details to make the main texts and supporting information clearer. The authors have also responded to the referees' comments point-by-point. While some of them are adequate, some of them are still unsatisfactory. In particular, the response "A time-resolved spectroscopic study of DMP⁺ in the gas phase would indeed be desirable but is clearly outside the scope of our work. We suspect that this may be a formidable challenge given the rarity of such studies in general, in particular in a time-resolved fashion. Clearly this must have been one of the reasons why the authors of refs. 7, 8 chose to look instead at the Rydberg state of neutral DMP." to my previous comment is not satisfactory in a few ways. Firstly and most importantly, the request for more and stronger experimental evidence of DMP⁺ is clearly within the scope because the manuscript claims to discover an unusual (and the first) case where Rydberg electron interacts with the cationic core so much that it alters the molecular structure in a fundamental way, which naturally requires a comparison between the Rydberg state and the cation. The fact that there are two interpretations with opposite opinions about the multi-reference character of the DMP⁺ system, which both seem to be able to explain the existing experimental observables in the references, emphasizes the importance of the direct measurement of gas-phase DMP⁺ cation. "This work presents the first case where interactions of a Rydberg electron with the underlying cationic core alter molecular structure in such a fundamental way" is a big claim if it is true, which definitely requires experimental support. Secondly, the authors of the

manuscript “suspect that the direct measurement of gas-phase DMP⁺ may be a formidable challenge” and “one reason the authors of refs. 7, 8 chose to look instead at the Rydberg state of neutral DMP”. This is not convincing to me. The Rydberg electron is widely and long-time known to have small interaction with thus negligible effect on the cationic core, so it is not surprising people would assume similar structures of them and use the Rydberg state to study the cation, as the Rydberg electron is easy to detach and used as a probe. While I agree that Rydberg state is more convenient to be used to study cation when it is appropriate (as in most cases), it is not formidable challenge to study the cation directly and should be carried out in order to prove the rarer cases when the previously mentioned scenario fails (as claimed in this manuscript, and why the manuscript is interesting in the first place). The gas-phase neutral DMP can be readily ionized by one VUV (~147nm) photon or two UV (~294nm) photons to generate DMP⁺ and probed by various methods such as structure-sensitive X-ray or electron scattering, or state-sensitive white light, or site-sensitive X-ray absorption or photoelectron spectroscopy. Without the necessary experimental support, the manuscript is an interesting alternative explanation of previous data, but not more.

We have to disagree fundamentally with the reviewer on this particular aspect. As pointed out in our previous response, an experimental confirmation of the absence of a localized minimum on the cation surface would be highly welcome, but it is not needed to establish the point beyond a reasonable doubt. The present calculations clearly demonstrate the difference between the absence of the localized minimum for the cation and its presence for the Rydberg state. Indeed, this has been shown by various different, adequate computational methods, matching even rather closely the levels used for DMP⁺ and DMP. It is simply not true that there are two different interpretations that equally explain the experimental observations. The results presented in this work clearly rule out the localized minimum on the cation surface and show that previous calculations that gave such a minimum were flawed due to unphysical discontinuities in the CASSCF wave functions and energies underlying the MRCI results. The fact that the Rydberg electron has such an unprecedented influence on the surface was indeed unexpected but is clearly supported by the data provided. There do not remain “two sides of the story” in this case.*

In addition, the connections to the previous experiments are still inadequate, or sometimes confusing. For example, even though the authors responded, “We do now point on p. 10 of the main text to LR-SCS-CC2 and IP-EOM-CCSD as particularly appropriate methods to treat the Rydberg-state surface, even though various TDDFT approaches also seem to provide accurate results.”, there are no direct comparison and discussion about the agreements with the experimental binding energies and relative energies for LR-SCS-CC2 and TDDFT. There is a short discussion of the EOM-CCSD results, but as this has been reported in reference 8 and the arXiv preprint reviewer 2 mentioned. In particular, according to the Supplementary Table 2, LR-SCS-CC2 underestimates the binding energy of the DMP-D* state and overestimates the binding energy of the DMP-L* state. It also overestimates the binding energy difference a lot. Most of the TDDFT results listed in the table are far from “accurate” comparing to the experimental data. On the other hand, the PZ-SIC results reported in reference 8 match the experiment much better. It is difficult to justify the choice of method when the method does not match the experimental data as well as a previously published work.

We do not agree that identical EOM-CCSD results have been reported in reference 8. While we give ionization energies at the IP-EOM-CCSD level based on RHF reference functions, reference 8 gave ground-state ionization energies from Δ CCSD(T) calculations augmented by

EOM-CCSD excitation energies. The calculated binding energies are actually quite close to ours, which might have created the confusion. This is most likely due to a) the fact that CCSD(T) mostly cures the artificial symmetry breaking in the underlying UHF reference, which is not fully eliminated at CCSD level, and b) that the Rydberg state and the cation actually share a lot of structural features. Notably, however, we follow the entire reaction path at IP-EOM-CCSD level, based on an optimized LR-SCS-CC2 path. This gives a much clearer picture than single points at minima obtained at MP2 level for the cation. And our EOM-CCSD calculations for cation and Rydberg state are both based on the same RHF reference for the neutral ground state, providing a particularly close comparison. We have added a discussion of these facts on p. 9 of the manuscript.

If the rather approximate, exotic PZ-SIC method applied to the cation gives a somewhat closer match with the experimental energy profile of the Rydberg state, this is a typical case of “the right answer for the wrong reason” and it does not provide an argument against the present results and line of reasoning. PZ-SIC has been shown many times to overcorrect GGA energies and wave functions dramatically, and we had given a number of references on this topic in the main text (p. 5). Notably, PZ-SIC or BHandHLYP are certainly no methods one would recommend to treat systems where one suspects appreciable static correlation, another contradiction in the previous work. It is important to understand that an apparent agreement between some approximately computed and experimental numbers can be highly deceiving, due to error compensation. The fact that IP-EOM-CCSD or LR-SCS-CC2 do not exactly match the experimental binding energies of the Rydberg electron does not change the fact that these are far more accurate methods than PZ-SIC. Notably, they provide qualitatively correct energy profiles for the Rydberg state that match the experimental observations (see also statement below). This also holds for the TDDFT calculations with various modern functionals, which show the agreement with the experimental binding energies one may expect at these levels. Only the deviations at TD- ω B97X-D level seem to be a bit large. EOM-CCSD is without doubt the most accurate method used so far to treat the Rydberg state, which is why we refrain from discussing the performance of the less accurate TDDFT results in detail.

Another two examples of confusing connections to the previous experiments are both on page 10. One is “The change in binding energies of the Rydberg electron also matches the observed experimental increase of about 0.11 eV from the localized to the delocalized state.”⁷ Apparently, the binding energy of the delocalized state is lower, not higher, than that of the localized state according to the cited reference 7.

The sentence had indeed been formulated incorrectly, it has now been corrected. Of course, the binding energy is larger for the localized state! We thank the reviewer for pointing this out.

The other one is “The LR-SCS-CC2/aug-cc-pVTZ energy of DMP-L* relative to DMP-D* (0.24 eV) underestimates somewhat the experimental estimate of 0.33 ± 0.04 eV obtained from thermal populations.”⁸ The 0.33 ± 0.04 eV in the cited reference 8 is the energy difference between the cations after considering the binding energy difference.

The reviewer is perfectly correct. This error has also been corrected, leading to a much better agreement between theory and experiment. The statement in the text now reads: “The LR-SCS-CC2/aug-cc-pVTZ energy of DMP-L relative to DMP-D* (0.24 eV) corresponds very well to the experimental enthalpy difference between the two states of 0.22*

+/- 0.04 eV.⁸ The improvement is somewhat reduced at the EOM-CCSD/aug-cc-pVTZ level (0.29 eV), hinting at error compensation effects in the SCS-CC2 treatment.” We have now also added experimental values to Supplementary Table 1, including the relatively rough kinetics-based estimate for the barrier.

There is one more confusing connection on page 12 of the supporting information. “but even at 980K structures similar to DMP-L+ a) do not account for a significant maximum in the combined distribution functions and b) are not observed immediately following ionization as suggested by experiments.³” How is this suggested by the experiments in the SI reference 3? The experiments apparently observed a localized state in the beginning at and shortly after t_0 .

Our formulation may not have been as clear as we thought. Our intention was to point out that the simulation for the cation does not produce the localized state right after ionization, while the Rydberg-state experiment does. We have now reformulated this statement to (SI, p. 10): “but even at 980 K structures similar to DMP-L⁺ do not account for a significant maximum in the combined distribution functions. Moreover, experiments suggest that these localized structures would need to be observed immediately following excitation,³ which is not the case in our simulations of the cation.”

In addition, the respond “This claim is contradicted clearly even by a basic chemical analysis of the system, as DMP exhibits a saturated sigma-framework. A large active space necessarily includes the C-N and C-C sigma-bonds in the active space (see below), which is completely unnecessary, as their contributions are excessively small in a final wave function of the radical cation and only add some dynamic and negligible static electron correlation.” does not consider that the C-C sigma* anti-bonds participate in the mixing with the nitrogen lone pairs to assist the charge delocalization procedure via the so called “through-bond-interaction”, as supported by the weakened elongated C-C bond in the X-ray scattering experiment in reference 9.

There seems to be a fundamental misunderstanding of the nature of multi-reference systems. The delocalization of the hole from the nitrogen lone pairs via the sigma bonds is undeniable as confirmed by the structural details pointed out by the referee. But this does not mean that the sigma-bonding framework is involved in substantial static correlation, which is made unlikely by the relatively different energies of the orbitals involved. Moreover, we find that the wave function of the cation is strongly dominated by one reference determinant (as pointed out on p. 6). This is true not only at CASSCF(11,12) but also at RASSCF(19,4,4; 9,1,10) level. This indicates that the delocalization via the sigma bonds and the mentioned structural distortions are not coupled to substantial static correlation and single-reference wave functions like, e.g., CCSD(T) or IP-EOM-CCSD describe these effects well.

There are also a few minor items that are needed to be improved. Firstly, some of the figures are hard to read. For example, Figure 1 has many curves in one plot, it would be easier to follow which is which with a proper legend of the curves and dots. The same suggestion goes to the figures in the SI too when there are many curves and dots. One can improve the readability of figures by changing the type of the curve and the shape of the plot symbols and using proper legend. Secondly, it would be clearer to mark which part or side of the plots is localized and which is delocalized. Some of the figures have the label but many do not. Thirdly, the real experiment is from the localized state to the delocalized state, it is a little counter-intuitive to have the reaction path reversed.

We have extended the legends of the plots and marked the localized and delocalized states in the figures. However, we think the selection of curve types is good for clarity as it is. We also prefer to keep the orientation of the plots, going from the more stable delocalized to the less stable localized side. Of course, this is ultimately a matter of taste.

In summary, this manuscript has interesting ideas that may change our way of thinking of Rydberg electrons and their effects on the molecular cationic cores. However, it lacks the critical experimental data to support these ideas. These data must be obtained before the conclusion can be drawn.

As pointed out above, we fundamentally disagree with the last two sentences of this comment but hope to have resolved all the other points of the referee.

REVIEWER COMMENTS

Reviewer #4 (Remarks to the Author):

Kaupp and co-authors address an interesting and controversial topic: whether a localized mixed-valent state of the dimethylpiperazine cation exists in a stable configuration or not. The authors employ state-of-the-art computational methods and conclude that a stable configuration does not exist. In doing so, they provide an explanation of why previously published calculations suggested otherwise. They further re-interpret the photoelectron spectroscopy results that initiated the debate, and demonstrate that the neutral Rydberg state presents a localized configuration, even though the cation does not. This last finding is striking and novel. The investigation performed by the authors is carefully conducted, as they tackle the problem with independent high-level methodologies, while considering alternative explanations, and presenting a collection of results that supports their conclusions. This contribution clearly unveils the problems of previous calculations and convincingly shows that the Rydberg electron can play a much unexpected role.

I think most of the concerns of reviewer #2 are not reasonable. Instead, the reviewer fails to recognize the merits of this work and the flaws of previous publications. I am compelled to agree with the authors and with reviewer #1 in the assessment that reviewer #2 seems to obfuscate the discussion. I also disagree with the statement of reviewer #3 concerning the need of experimental data to support the conclusion of the authors regarding the active role of the Rydberg electron. While experimental results would certainly be desirable, Kaupp and co-workers present strong theoretical evidence that supports their conclusion.

I will be happy to recommend publication in Nature Communications after the authors clarify the points raised below. I also have a few recommendations on complementary calculations which, I believe, could make the authors' arguments even more compelling thus strengthening the manuscript.

1) The calculations for the cation reveal a relatively flat curve at the localized region. It is not unreasonable that higher-order dynamical correlation effects or subtle geometrical effects could come into play and create the conditions for binding the localized cation. In particular, the difference between CCSD and CCSD(T) results are significant and clearly show that triple excitations are relevant, hinting at important higher-order correlation effects. To more convincingly rule out the possibility of a minimum for the localized state, the authors could run CC3 (and/or IP-EOM-CC3) calculations at few key points in their curves, a more accurate method than CCSD and considerably less expensive than CCSDT. They could also perform CCSD and CCSD(T) calculations for a ROHF reference wave function, which would eliminate the spin contamination problem of the UHF reference wave function. Based on the authors' argument, the (T) contribution would be expected to be smaller in this case. Furthermore, it is possible that optimizing the geometry at a more accurate level of theory than BHandHLYP (which has its issues as correctly pointed out by the authors) could produce a stable minimum. If it does not appear for a ROHF-CCSD optimization, this would further strengthen the argument that the UHF-CCSD minimum would be an artifact. Otherwise, the possibility for a minimum should remain open. Notice that this discussion concerns the possible existence of a very shallow minimum, and does not change the correct identification of problems in the previous calculations and most likely does not change the conclusion about the Rydberg electron stabilizing the cation. These are mostly suggestions which I think would provide stronger evidence for the authors' claim on the absence of a stable localized state of the cation. If the authors decide not to proceed with additional calculations, they should modify the manuscript accordingly and acknowledge that such a shallow minimum may exist, even if unlikely, in light of the observed flat curve and the above mentioned arguments.

2) The state-specific and state-averaged CASSCF calculations, which account respectively for one and two states, show qualitatively different potential energy curves. One could extend the argument and ask whether the latter curve could show artificial features because other low-lying excited states are not included in the state-averaging. Have the authors considered that? Based on

the three close-lying UHF solutions (two localized and one delocalized), including three states (as done for the NOCI calculation) in the state-averaging would be more natural than selecting only two. I think it is important to further elaborate on this aspect, mentioning the character of the excited states and showing their corresponding curves.

3) I am puzzled by the connection made between the observed discontinuities and what the authors refer to as "symmetry-breaking". Assuming they refer to a spin symmetry breaking, they should quantify the spin contamination along the potential energy curves. However, the three UHF solutions behave continuously along the pathway, which do not indicate the presence of Coulson-Fischer points associated with spin symmetry breaking. In this sense, the spin symmetry would be broken along the whole path, and the discontinuities would have nothing to do with a symmetry breaking. Rather, they would simply reflect different UHF solutions that cross in energy, some of which could be artificially stabilized due to spin symmetry breaking. A similar analysis of the ROHF solutions would help to clarify this matter. The usage of the term "symmetry-breaking" becomes more problematic in the context of CASSCF. There should be no spin contamination for a spin-restricted CASSCF wave function. It is thus not clear what "symmetry breaking" is alluded to in this part of the discussion, which should be revised.

4) The authors state that "All total electronic energies generated in this study are provided in the main text and Supplementary Information." I could not find them in either one. For reproducibility, it is important to include the total energies. I also suggest adding a figure to the Supplementary Information comparing their different curves for the cation obtained with high-level calculations. This would be helpful to illustrate that different theoretical approaches produce the same behavior. In the Supplementary Information, the meaning of the sentence "...which results in more physically correct states that, for example, avoid crossings" is unclear. The caption of Figures 1 and 2 says eV, but kJ/mol is used in both figures. Please correct.

Referee #4

Kaupp and co-authors address an interesting and controversial topic: whether a localized mixed-valent state of the dimethylpiperazine cation exists in a stable configuration or not. The authors employ state-of-the-art computational methods and conclude that a stable configuration does not exist. In doing so, they provide an explanation of why previously published calculations suggested otherwise. They further re-interpret the photoelectron spectroscopy results that initiated the debate, and demonstrate that the neutral Rydberg state presents a localized configuration, even though the cation does not. This last finding is striking and novel. The investigation performed by the authors is carefully conducted, as they tackle the problem with independent high-level methodologies, while considering alternative explanations, and presenting a collection of results that supports their conclusions. This contribution clearly unveils the problems of previous calculations and convincingly shows that the Rydberg electron can play a much unexpected role.

I think most of the concerns of reviewer #2 are not reasonable. Instead, the reviewer fails to recognize the merits of this work and the flaws of previous publications. I am compelled to agree with the authors and with reviewer #1 in the assessment that reviewer #2 seems to obfuscate the discussion. I also disagree with the statement of reviewer #3 concerning the need of experimental data to support the conclusion of the authors regarding the active role of the Rydberg electron. While experimental results would certainly be desirable, Kaupp and co-workers present strong theoretical evidence that supports their conclusion.

We thank the reviewer for the positive assessment.

I will be happy to recommend publication in Nature Communications after the authors clarify the points raised below. I also have a few recommendations on complementary calculations which, I believe, could make the authors' arguments even more compelling thus strengthening the manuscript.

1) The calculations for the cation reveal a relatively flat curve at the localized region. It is not unreasonable that higher-order dynamical correlation effects or subtle geometrical effects could come into play and create the conditions for binding the localized cation. In particular, the difference between CCSD and CCSD(T) results are significant and clearly show that triple excitations are relevant, hinting at important higher-order correlation effects.

To more convincingly rule out the possibility of a minimum for the localized state, the authors could run CC3 (and/or IP-EOM-CC3) calculations at few key points in their curves, a more accurate method than CCSD and considerably less expensive than CCSDT.

While we agree that an IP-EOM-CC3 calculation would be impressive, there does not seem to be an efficient, parallelized implementation, which would enable us to apply this method to DMP⁺. Regular CC3 calculations will likely not add much further insight, as they are not

consistently better than CCSD(T) (as stated already when the approach was introduced, DOI: 10.1063/1.473322). Additionally, CC3 calculations on open-shell systems are a formidable task, as most codes (eT, Dalton, CFOUR) either do not allow open-shell calculations at all or have not parallelized the respective parts of the code. As stated in our reply on the first set of referee comments, Dr. Bryan Wong from UC Riverside has informed us that he managed to run CCSDT calculations and finds exactly the same curve as for CCSD(T). With the computational resources available to us, we have not been able to complete CCSDT computations in a reasonable basis set.

They could also perform CCSD and CCSD(T) calculations for a ROHF reference wave function, which would eliminate the spin contamination problem of the UHF reference wave function. Based on the authors' argument, the (T) contribution would be expected to be smaller in this case.

We have performed additional calculations based on ROHF reference wave functions. This results in virtually identical results as obtained with a UHF reference. We now give the additional results in the SI, pp. 3-5. We would, however, not expect the (T) contributions to be smaller in this case. This may be a misunderstanding due to an insufficiently clear wording used by us in the paper regarding the symmetry breaking (see below).

Furthermore, it is possible that optimizing the geometry at a more accurate level of theory than BHandHLYP (which has its issues as correctly pointed out by the authors) could produce a stable minimum. If it does not appear for a ROHF-CCSD optimization, this would further strengthen the argument that the UHF-CCSD minimum would be an artifact. Otherwise, the possibility for a minimum should remain open.

Optimizing the structure at the ROHF-CCSD level would not be sufficient, as the barrier still exists at this level (see Supp. Fig. 4). A full optimization of the reaction path at any coupled-cluster level beyond CC2 would in any case be extremely difficult (due to the open-shell nature of the system).

Notice that this discussion concerns the possible existence of a very shallow minimum, and does not change the correct identification of problems in the previous calculations and most likely does not change the conclusion about the Rydberg electron stabilizing the cation. These are mostly suggestions which I think would provide stronger evidence for the authors' claim on the absence of a stable localized state of the cation. If the authors decide not to proceed with additional calculations, they should modify the manuscript accordingly and acknowledge that such a shallow minimum may exist, even if unlikely, in light of the observed flat curve and the above mentioned arguments.

Based on the additional calculations provided, we are even more certain than before that such a minimum in fact does not exist. While it is basically impossible to completely disprove the existence of any species, we are confident that our diverse high-level calculations (which match the results from most commonly applied DFT methods) make the existence of such a minimum extremely unlikely. This is supported further by Wong's CCSDT results mentioned above. We have, however, now modified our wording to keep the remote possibility open that such a

minimum might exist. That is, on pp. 6/7 we now state "...that it is extremely unlikely that DMP⁺ exhibits a local DMP-L⁺-type minimum!"

2) The state-specific and state-averaged CASSCF calculations, which account respectively for one and two states, show qualitatively different potential energy curves. One could extend the argument and ask whether the latter curve could show artificial features because other low-lying excited states are not included in the state-averaging. Have the authors considered that? Based on the three close-lying UHF solutions (two localized and one delocalized), including three states (as done for the NOCI calculation) in the state-averaging would be more natural than selecting only two. I think it is important to further elaborate on this aspect, mentioning the character of the excited states and showing their corresponding curves.

This is a very reasonable consideration. We have now included additional data (Supp. Fig. 13) showing clearly that the inclusion of the second excited state does not change our results. This is due to the fact that this state is almost 6 eV higher in energy than the ground state (according to SA-CASSCF). The curves remain thus closely similar.

3) I am puzzled by the connection made between the observed discontinuities and what the authors refer to as "symmetry-breaking". Assuming they refer to a spin symmetry breaking, they should quantify the spin contamination along the potential energy curves. However, the three UHF solutions behave continuously along the pathway, which do not indicate the presence of Coulson-Fischer points associated with spin symmetry breaking. In this sense, the spin symmetry would be broken along the whole path, and the discontinuities would have nothing to do with a symmetry breaking. Rather, they would simply reflect different UHF solutions that cross in energy, some of which could be artificially stabilized due to spin symmetry breaking. A similar analysis of the ROHF solutions would help to clarify this matter. The usage of the term "symmetry-breaking" becomes more problematic in the context of CASSCF. There should be no spin contamination for a spin-restricted CASSCF wave function. It is thus not clear what "symmetry breaking" is alluded to in this part of the discussion, which should be revised.

We may have been less clear in our formulations than we thought. We do not refer to the breaking of spin symmetry but to the breaking of spatial symmetry as in the famous allyl radical example. For the symmetric DMP-D^{+/} minimum, both UHF and ROHF prefer a solution that does not belong to an irreducible representation of the molecular (nuclear) point group due to an overstabilization of a localized hole. Also, both HF and CASSCF break the symmetry present in the orbitals at some point along the reaction coordinate, leading to the cusp-like "barriers" in all these methods. It is this symmetry breaking that we refer to throughout the manuscript. Spin-symmetry breaking in the UHF wave functions is indeed only minor as can be seen from the newly added results using ROHF references (see above). We have further clarified our statements throughout the manuscript and in the SI (pp. 5, 6, 12, S3, S7) by generally adding "spatial" to the symmetry-breaking argument.*

4) The authors state that "All total electronic energies generated in this study are provided in the main text and Supplementary Information." I could not find them in either one. For reproducibility, it is important to include the total energies.

The total energies have been provided in a separate .xls file that seems to not have been correctly uploaded in the last submission. We adjusted the statement in the text to make clear that total energies are only found in the SI.

I also suggest adding a figure to the Supplementary Information comparing their different curves for the cation obtained with high-level calculations. This would be helpful to illustrate that different theoretical approaches produce the same behavior.

We had considered such an illustration before but as most methods provide extremely similar, simple behavior, this becomes too crowded to be informative, and we had decided against it.

In the Supplementary Information, the meaning of the sentence "...which results in more physically correct states that, for example, avoid crossings" is unclear.

We have now changed the statement to "...which corrects the unphysical state crossing in the HF solutions."

The caption of Figures 1 and 2 says eV, but kJ/mol is used in both figures. Please correct.

We have corrected the error in both figures.

REVIEWERS' COMMENTS

Reviewer #4 (Remarks to the Author):

The authors have fully addressed my questions. They clarified the few minor points that remained and also provided additional results that strengthen their conclusion. This contribution brings closure to a hotly debated topic, and I therefore recommend its publication.